# Flood drivers and trends: a case study of the Geul River Catchment (the Netherlands) over the past half century

Athanasios Tsiokanos[1,2], Martine Rutten[1], Ruud J. van der Ent[1], and Remko Uijlenhoet[1]

[1]Department of Water Management, Faculty of Civil Engineering and Geosciences, Delft University of Technology, Delft, The Netherlands
[2]Department of Operational Water Management and Early Warning, Unit of Inland Water Systems, Deltares, Delft, The Netherlands

**Correspondence:** Athanasios Tsiokanos (Athanasios.Tsiokanos@deltares.nl)

**Abstract.** Extreme precipitation in July 2021 caused devastating flooding in Germany, Belgium and the Netherlands, particularly in the Geul river catchment. Such precipitation extremes were not previously recorded and were not expected to occur in summer. This contributed to poor flood forecast and hence to large damage. Climate change was mentioned as a potential explanation for these unprecedented events. Yet, before such a statement can be made, we need a better understanding of the drivers of floods in the Geul and their long-term variability, which are poorly understood and have not been examined recently. In this paper, we use an event-based approach to identify the dominant flood drivers in the Geul and employ a multi-temporal trend analysis to investigate their temporal variabilities, as well as, a novel methodology to detect the dominant direction of a trend. Results suggest that extreme 24-hour precipitation alone is typically insufficient to cause floods. The joint probability of extreme and prolonged rainfall combined with wet initial conditions (compound event) determines the chances of flooding. Flood producing precipitation shows a consistent increase in the winter half-year, a period in which more than 70% of extremely high flows have occurred historically. While no consistent trend patterns are evident in the majority of precipitation and extreme flow trends in the summer half-year, an increasing direction in the recent past is visible.

## 1 Introduction

In July 2021, Western Europe was struck by extreme precipitation, leading to disastrous flooding in Germany, Belgium, and the Netherlands (Journée et al., 2023; Kreienkamp et al., 2021). This event ranked among the most devastating natural disasters to hit Europe in the past 50 years, resulting in at least 220 deaths and causing an estimated economic damage of approximately EUR 46 billion (MunichRe, 2022; Mohr et al., 2023). In the Netherlands, the Geul river catchment (344 km$^2$) was the most impacted, where the economic damage of the floods was estimated to exceed EUR 200 million, constituting approximately 50% of the total estimated damage in the country (Task Force Fact Finding hoogwater 2021, 2021). The event revealed weaknesses in flood risk management. Flood risks were poorly communicated to the inhabitants (Slager, 2023). The flood forecasting system for the Geul was under maintenance, and even if it had been operational, predictions would not have been accurate, because of poor representation of flood-generating processes, according to the responsible authorities (Task Force Fact Finding hoogwater 2021, 2021). The Geul catchment is considered quite an exceptional and atypical catchment for the Netherlands due

to its steep topography (elevations range from 40 to nearly 400 m) and deep soils (tens of meters). A proper understanding of flood drivers in the area is considered an important stepping-stone in mitigating the risks in the future.

Floods in a catchment are caused by the interaction of meteorological, river system, and catchment characteristics (Andrés-Doménech et al., 2015). Hydrological catchment properties can regulate streamflow response (Sharma et al., 2018), for example, extreme precipitation does not always result in floods in various basins around the world (Wasko and Nathan, 2019; Nanditha and Mishra, 2022; Berghuijs et al., 2019). Among the catchment characteristics, antecedent conditions (e.g., soil moisture) can play a crucial role in driving high flows (Bertola et al., 2020; Woldemeskel and Sharma, 2016). Many locations around the world have experienced the effects of wet antecedent conditions on flood risk (e.g., Garg and Mishra, 2019; Bischiniotis et al., 2018; Ivancic and Shaw, 2015; Cao et al., 2019). Especially in lowland catchments the discharge response can be strongly influenced by the catchment wetness, due to shallow groundwater and its effects on rainfall flow paths (Brauer et al., 2018). As a result, determining the relative contribution of antecedent wetness conditions and extreme precipitation in causing high river flows is critical.

The identification of the drivers of observed flood events has received increasing attention in the recent literature (e.g., Blöschl et al., 2019; Bertola et al., 2020). Examining relations between trends or seasonality in flood peaks and factors such as extreme precipitation or soil moisture to define flood drivers is well established (Blöschl et al., 2019; Do et al., 2017; Tramblay et al., 2021; Wasko et al., 2020). However, to enhance our knowledge of flood dynamics, an event-based approach has been suggested (Nanditha and Mishra, 2022; Tramblay et al., 2021; Berghuijs et al., 2019). This approach entails identifying the specific drivers behind individual flood events or extremely high-flow occurrences. By analyzing the characteristics and circumstances surrounding these events, one can gain valuable insights into the mechanisms and factors that contribute to their intensity and occurrence (Nanditha and Mishra, 2022).

A proper understanding of flood drivers does not only include their identification but also their long-term change. Investigating the changes of variables that can cause a hazard is crucial for managing the risks in an effective way (Yang et al., 2021) and can facilitate the planning of reliable and meaningful interventions. Making a critical assessment of the past and current states and providing the long-term trends of hydroclimatic variables play a key role in future projections (Squintu et al., 2021). Trends and changes in the time series of hydrological and climatological data have received attention in catchments and areas across the world (e.g., Blöschl et al., 2017; Do et al., 2017; Hannaford et al., 2021; Murphy et al., 2020).

However, most of the existing trend tests are limited, as they are conducted within fixed timeframes, which may fail to accurately capture the historical variability. The significance and magnitude of trends can vary considerably based on the chosen study period and duration. To deal with this limitation, multi-temporal trend approaches have been leveraged (e.g., Hannaford et al., 2021; Murphy et al., 2020), considering all possible combinations of start and end-year periods. Although these analyses have helped in identifying temporal variabilities, a research gap remains in determining the main trend direction, such as consistency or stability, across all studied time frames. Lupikasza (2010) developed criteria for expressing trend stabilities using a fixed 30-year moving window, however, this approach did not fully consider the entire variability (multi-temporal analysis) but instead utilized an overlapping period which can be misleading. Since each 30-year window overlaps with the previous and subsequent windows, trends can be missed or misinterpreted because the overlapping periods could obscure them.

Using only overlapping periods can lead to artificially smooth trends not representing the true (long-term) variability of the
data. In addition, the length of the selected moving window might introduce bias in the analysis. Trends over longer periods
and with different combinations of start and end-year periods expressing the full historical variation should also be taken into
account in the calculation of the main direction. To address these limitations, our study builds on the multi-temporal approach
and develops a methodology capable of identifying and assessing trend consistency in multi-temporal analyses, taking into
account the complete range of variability. This new method is anticipated to deepen our understanding of flood driver trends in
the Geul River catchment, with potential applicability across broader contexts.

In summary, the Geul river catchment consists an interesting example of a hilly catchment in northwest Europe, with unique
hydrological characteristics that are shaped by the underlying geology, topography and land use. The recent floods have shown
that there is a need for further research into the drivers of flooding in the area, particularly in the context of climate change.
Specifically, the role of extreme precipitation and antecedent conditions as potential flood drivers and their long-term variability
remain to be examined. Therefore, our objective is to detect the primary drivers of high-flow/flood events in the Geul River
catchment and analyze their long-term trends. To achieve these objectives, we address the following scientific questions that
are crucial for our understanding of floods in the Geul River catchment: (a) *What are the dominant flood drivers in the Geul
river catchment in different seasons?* and (b) *What are the trends (temporal variability and consistency) of the flood producing
precipitation and extreme discharges in the catchment?* To identify the dominant flood drivers we use an event-based approach.
In addition, we utilize a multi-temporal trend analysis to investigate the temporal variabilities of the trends and introduce a new
methodology to detect the dominant direction (i.e. consistency) of a trend. Although our study focuses on the Geul area, it is
essential to highlight that our combined approaches (integrating an event-based approach with multi-temporal analyses) and
proposed trend consistency method hold applicability beyond this specific case. Thus, our aim is to offer valuable insights for
the Geul area while avoiding constraining the scope of our methods and findings to a singular case study.

## 2 Methods and materials

### 2.1 Study area

The Geul river is an important tributary of the Meuse and is located in the Netherlands, Belgium, and part of Germany, close
to the three-border region (Fig. 1). The total area of the Geul catchment is approximately 344 km$^2$. The Geul drops about
250 m over approximately 60 km, making it one of the few steeply sloping rivers in the Netherlands. The Geul river has an
average discharge of approximately 3.2 m$^3$s$^{-1}$ (at the outlet of the catchment) and is mainly rain-fed. As a consequence, its
discharge can change dramatically during flood and drought events (e.g. ranging from 1 m$^3$s$^{-1}$ during drought periods to more
than 40 m$^3$s$^{-1}$ during floods). The response time of the catchment is in the order of one to two days (Asselman et al., 2022).
The annual average precipitation is approximately 870 mm yr$^{-1}$ and is rather uniformly distributed over the year (see Fig.
2a). Average annual discharge at the outlet of the catchment and potential evaporation are about 307 mm yr$^{-1}$ and 585 mm
yr$^{-1}$ respectively, based on time series from 1970 to 2021. The flow regimes in the Geul do not show large variations (Fig. 2).
The, on average, rather equal distribution of runoff in the Geul throughout the year is due to the effect of groundwater storage

provided by large chalk aquifers in the catchment (Tu, 2006). The long-term evaporative index is approximately 0.67 and the runoff ratio 0.35.

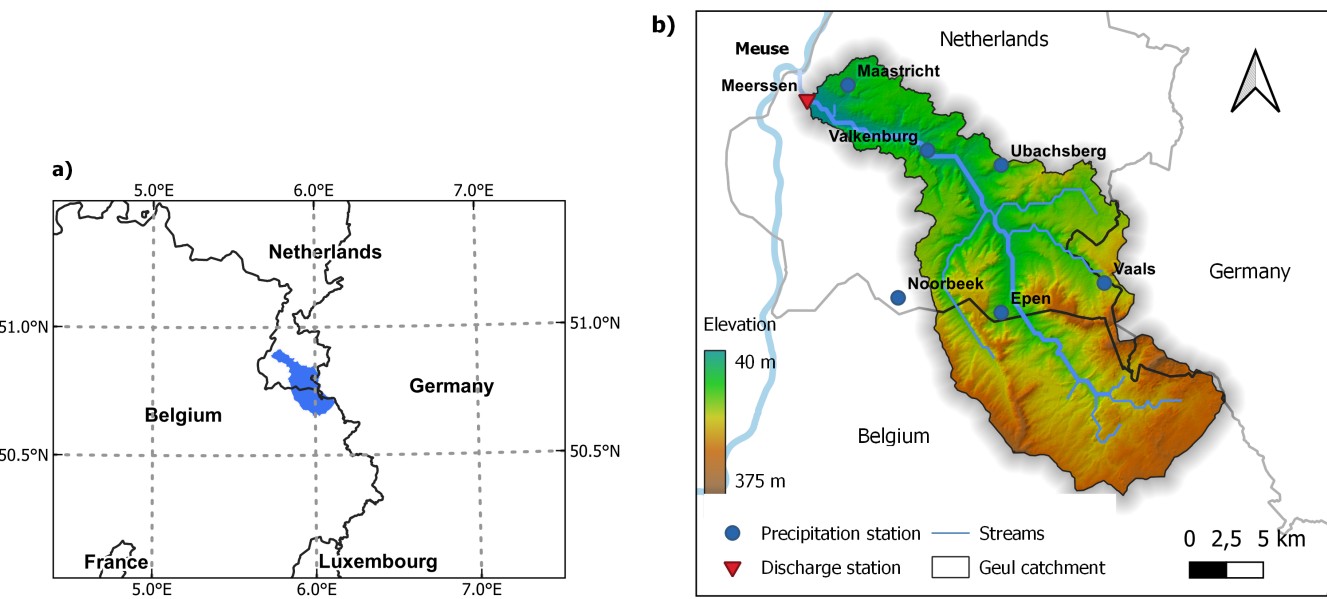

**Figure 1.** (a) Location of the Geul river catchment (blue shaded area), (b) elevation map of the study catchment including the location of precipitation stations and the discharge observation station at the outlet.

## 2.2 Data sets

Streamflow time series at the outlet of the catchment (station Meerssen), were made available by the local water authority Waterschap Limburg, including 15-minute measurements from 1970 up to August 2021. From 1970 to 2011 measurements were taken using a measuring weir, while since 2011 discharges are measured using Acoustic Doppler Current Profilers (ADCP) (van der Deijl, 2023). Historical flood event data (i.e. date of occurrence) are based on Thewissen (2022), who performed a local newspaper search using Delpher, a Dutch database containing digitized texts from newspapers, books, and magazines.

These collections are curated by scientific organizations, libraries, and heritage institutions. Thewissen (2022) obtained the data through an iterative process that involved Optical Character Recognition (OCR) and manual scanning. In this research, a flood is defined as surpassing the bankful capacity. For detailed information about the search methodology, please refer to Thewissen (2022).

Records of 24-hour precipitation from five stations located in (or near) the Geul river catchment are used (see Fig. 1).

The data used in this study come from the Royal Netherlands Meteorological Institute (KNMI) manual rain gauge network. Volunteer observers operate the rain gauges on a 24-hour basis (from 8 am to 8 am local time). In addition to the volunteer KNMI stations, daily (calendar days) measurements from the automated meteorological station at Maastricht are used. Data from automated stations are also available at an hourly basis, however the daily scale is used in order to have the same resolution

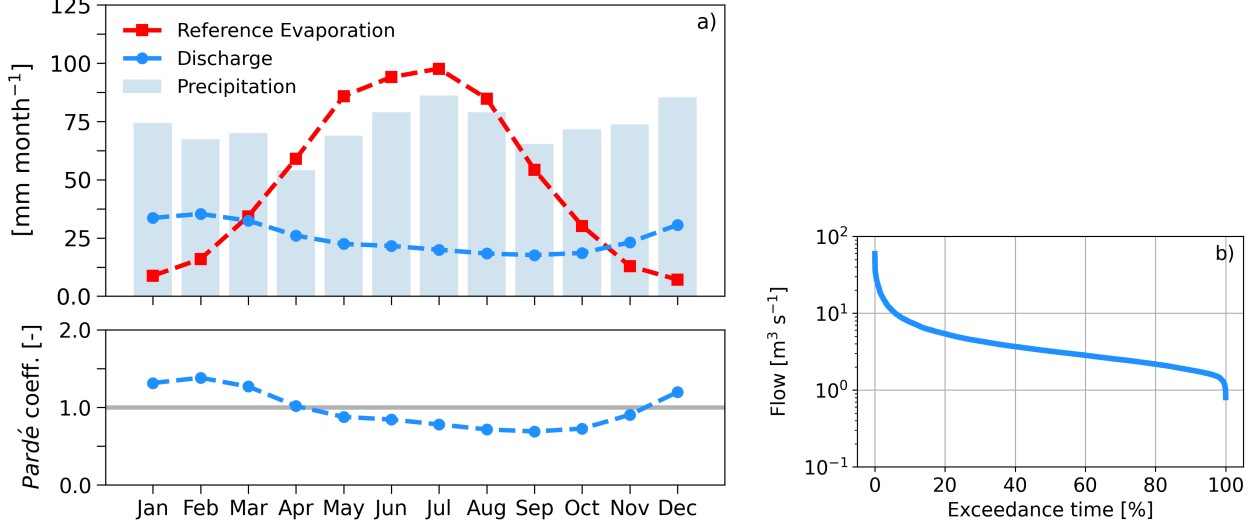

**Figure 2.** (a) Upper panel: Monthly averages for precipitation (averaged using Thiessen polygons), reference (Makkink) evaporation (obtained from the Maastricht meteorological station) and discharge in the Geul catchment, Lower panel: Flow regimes of the Geul based on Pardé coefficient, defined as the ratio of the average monthly discharge to the average annual discharge, (b) Flow duration curve along the Geul based on the mean daily discharge. All graphs are produced using time series from 1970 to 2021.

for each station. The time series are complete (except for three missing months at Noorbeek) and are considered to be of high
quality, as KNMI performs regular quality tests (Buishand et al., 2013). Days with extremes were checked and it was found
that all of them occurred during well-known high precipitation or flood events.

The available precipitation record periods slightly differ between the selected stations and mainly start in the 1950s, except
the new Epen station, which has a record from 1981 onward. Specifically, stations at Vaals, Valkenburg, and Noorbeek have a
record from 1951 onward, while the available time series at the Maastricht and Ubachsberg stations start from 1958 and 1955,
respectively. Due to its shorter available period the station Epen is excluded from the analysis. The precipitation stations located
in the uppermost regions of the catchment in Belgium were not taken into account due to their limited temporal coverage, lack
of alignment with the discharge (and the KNMI) data, and unsuitability for trend analysis.

Another important factor to consider when analyzing meteorological time series, and especially trends, is data (in)homogeneity.
Trends represent the long-term changes in the data over time caused by natural/climate variability, while inhomogeneities rep-
resent changes in the data that are not part of the underlying trend. There are several types of modifications that can occur and
can cause significant non-climatic alternations (inhomogeneities) in the data, especially in long time series, such as changes
in the location of the measuring station, differences in the manner and the procedures (e.g. measuring frequency) or changes
in the instruments/tools. Performing statistical tests for homogeneity on precipitation data measured at daily frequencies is
challenging or practically infeasible (Lupikasza, 2010). This can be attributed to the fact that daily time series show strong
random variations. For this reason, the precipitation time series were aggregated to monthly sums and then tested using two

well-known homogeneity tests: the Standard Normalised Homogeneity Test (SNHT) (Alexandersson, 1986; Alexandersson and Moberg, 1997) and Buishand's U test (Buishand, 1984). Both tests were applied at a significance level of 5% ($\alpha$=0.05). The test results showed that the data time series were free of significant errors and no inhomogeneities were detected.

## 2.3 Flood driver detection

### 2.3.1 Extreme indicators

Hydrological years are used instead of calendar years to ensure a more accurate representation of the frequency and intensity of extreme events that may occur within a particular hydrological regime. Similarly to Tu (2006) and Sperna-Weiland et al. (2015), we define a hydrological year from November to October, for the study area. Half-year hydrological winter (from November to April) and summer (from May to October) are also considered.

Two types of flood data time series are used: annual maxima (denoted as $Q_{max}$), including winter and summer yearly maxima (denoted as $Q_{W,max}$ and $Q_{S,max}$, respectively), and extremely high flows using the peaks over threshold (POT) method (Haan, 2002). The maximum daily discharge is extracted from the 15-min discharge data and hydrological years with over 20% missing daily values are omitted from the analysis. Consequently, the years 1971, 1974, and 1990 have been excluded from the $Q_{max}$ times series. For identifying extremely high flows we use the 99th percentile threshold (Nanditha and Mishra, 2022; denoted as $Q_{99}$) extracted from the maximum daily discharge time series from 1970 to 2021, excluding daily missing values. We use only extremely high events separated by a time frame of five days to ensure that the selected high flows are independent and do not belong to the same flood wave.

Floods are generally caused by a combination of initial moisture conditions and precipitation. To explore the probable causes of high flow episodes, we employ six indicators to assess precipitation and antecedent soil moisture levels: extreme precipitation (denoted as $P_{99}$), multi-day precipitation (denoted as $P_{MD}$), wet antecedent conditions (denoted as $P_{WAC}$), $P_{99}$ combined with $P_{WAC}$ (denoted as Compound I), $P_{MD}$ combined with $P_{WAC}$ (denoted as Compound II), and $P_{99}$ and $P_{MD}$ combined with $P_{WAC}$ (denoted as Compound III). These indicators allow us to examine the relative role of extreme precipitation, prolonged heavy rainfall, extreme soil moisture conditions, and compound extremes in generating high flows.

We estimate $P_{99}$ as the events that exceed the 99th percentile of wet days (days with more than 1 mm precipitation) (Nanditha and Mishra, 2022). We define $P_{MD}$ events using the 95th percentile of all $k$-day accumulated (rolling sum) precipitation time series (Nanditha and Mishra, 2022). To clarify the $P_{MD}$ definition, we ensure that the 95th percentile of multi-day rainfall consistently surpasses the 99th percentile of the 24-hour rainfall on wet days, aiding in distinguishing between $P_{99}$ and $P_{MD}$. In this way usually more than two days of precipitation are necessary to exceed the $k$-day 95th percentile and trigger $P_{MD}$, allowing the assumption that $P_{MD}$ can be used as a proxy of heavy prolonged events. As we use the 95th percentile of all $k$-day accumulated (rolling sum) precipitation to define $P_{MD}$ and we have "daily" values, this threshold is expected to be exceeded in prolonged events irrespective of the selected duration, indicating that we have prolonged (multi-day) heavy events (larger than the 95th percentile of the selected $k$-day accumulations), although not so extreme as the 24-hour $P_{99}$, which helps us examine the relative contributions of extreme precipitation and prolonged heavy rainfall in generating high flows. However, in rare cases,

24-hour precipitation can simultaneously trigger both $P_{99}$ and $P_{MD}$ especially for the lower $k$-day accumulation periods, which is unavoidable. Thus, for each of the five precipitation stations considered, we calculated the $P_{MD}$ 95th percentile for different durations. It was found that a duration longer than 4 days is required for this percentile to surpass the 99th percentile used in defining $P_{99}$. Finally, to determine the most suitable $k$-day $P_{MD}$ duration for $k \geq 4$, we evaluate the $P_{MD}$ probability preceding high flows across multi-day precipitation durations up to 10 days (see Sec. 3.1.1).

Furthermore, we use the Antecedent Precipitation Index (ratio of 30-day pre-event precipitation to long-term average for the same period; API) as developed by Marchi et al. (2010), to access the initial catchment conditions and get an estimate of the initial (soil) conditions. Marchi et al. (2010) classifies antecedent moisture conditions as follows: (1) dry: $0 < \text{API} \leq 0.5$; (2) normal: $0.5 < \text{API} \leq 1.5$; and (3) wet: $\text{API} > 1.5$. $P_{WAC}$ corresponds to API values higher than 1.5 (Marchi et al., 2010). The API's effectiveness in assessing initial soil wetness conditions was documented for instance by Marchi et al. (2010), who demonstrated its strong agreement with predictions from a continuous soil moisture accounting hydrological model (Norbiato et al., 2008). However, since the index is based solely on precipitation, its sensitivity to evaporation is further discussed in Sect. 3.1.5. This is done by computing the 30-day pre-event effective rainfall, which entails subtracting reference evaporation obtained from the Maastricht station from the precipitation measurements.

### 2.3.2 Monthly distribution of extremes

In order to get a rough indication of the effects of extreme rainfall on high flows, we estimate the monthly distribution of $P_{99}$, $P_{MD}$, annual maximum precipitation (denoted as $P_{max}$) and high flow extremes (i.e. $Q_{99}$ and $Q_{max}$), together with the past flood events (as defined in Sect. 2.2). Despite the fact that the monthly distribution of $P_{WAC}$ events cannot be directly related to the monthly distribution of high flow extremes, as $P_{WAC}$ indicate wetter than average conditions any time of the year, their relative frequencies can provide useful insights. For this reason, we also calculate the monthly frequencies of $P_{WAC}$. To achieve this each, daily timestep in the precipitation series is treated as an individual event. This involves summing the precipitation amount for each day over the previous 30 days and then dividing this sum by the 30-day long-term average for the same period across the entire time series. In this way, an API index per day is obtained.

The use of all-day percentiles for discharge ($Q_{99}$) and wet-day percentiles for rainfall ($P_{99}$) may lack statistical robustness, primarily because there could be a potential increase in the number of wet days over the study period (Schär et al., 2016). To address the potential issue a trend test in the number of wet days per year and station was conducted using the non-parametric Mann-Kendall (M-K) (Kendall, 1955; Mann, 1945) test at a significance level of 0.05. The analysis did not reveal any significant trends in the number of wet days.

### 2.3.3 Event-based analysis

After the seasonal assessment of the extreme indicators and high-flow events, an event-based approach is followed to detect the predominant flood driver(s) in the catchment. Specifically, we calculate the likelihood of the six precipitation indicators (i.e., $P_{99}$, $P_{MD}$, $P_{WAC}$, Compound I, Compound II and Compound III) preceding selected high flow episodes (i.e. $Q_{max}$) in order to determine the primary flood cause. We use the $Q_{max}$ events instead of the $Q_{99}$, as they are better defined for identifying

flood drivers including all major flood events, considering also that the $P_{99}$ is calculated from a significantly longer data series than the $Q_{99}$. We conduct a comprehensive analysis of each high-flow event, examining the frequency of occurrence for each indicator before a $Q_{\max}$ event. Specifically, we determine how many times an indicator occurred in relation to the total number

of $Q_{\max}$ events, effectively calculating its likelihood in contributing to such occurrences. For instance, if $P_{99}$ is the main flood cause in the catchment, then $P_{99}$ is highly likely to occur during the duration of the flood (precipitation amount of the same day) or within one day before the commencement of the event. Here, we use a two-day interval (i.e. precipitation at the same or the previous day of the flood event) for mainly two reasons: (a) the lag time of the catchment and (b) to eliminate errors due to the use of daily values (e.g., a high flow event may occur early morning due to $P_{99}$ from the previous day). For Compound

I events ($P_{99}$ on $P_{\mathrm{WAC}}$), we verify whether the $P_{99}$ on $P_{\mathrm{WAC}}$ occurred either on the event day itself or if the $P_{99}$ on $P_{\mathrm{WAC}}$ took place one day prior to the event. This approach guarantees that the $P_{99}$ consistently occurs in pre-existing wet conditions. Thus, we establish the requirement for a $P_{99}$ to appear on $P_{\mathrm{WAC}}$ for a compound event, thereby preventing scenarios where a $P_{99}$ occurring one day prior to the $Q_{\max}$ under normal circumstances could increase the API on the day of the $Q_{\max}$, leading to a $P_{\mathrm{WAC}}$. In a similar manner, we calculate the probability of Compound III preceding the $Q_{\max}$ events: $P_{\mathrm{MD}}$ and $P_{99}$ and

$P_{\mathrm{WAC}}$ on the day of the event or one day before. For the remaining indices, $P_{\mathrm{MD}}$ and $P_{\mathrm{WAC}}$, we simply verify whether these indicators are present on the day of the event.

As we directly compare the likelihood of each indicator leading to $Q_{\max}$, it is important to note that the comparison among different flood drivers may be affected by the rarity of certain events. The total number of flood indicators varies, particularly for the most extreme ones, such as $P_{99}$, Compound I, and Compound III. For instance, in some years, a $P_{99}$ event might not

occur, and therefore, it cannot precede $Q_{\max}$. To account for this effect, the reverse scenario is also considered: given that a driver has occurred, what is the relative frequency of it being followed by a $Q_{\max}$ event. Thus, we calculate the number of unique hydrological years in which a driver is observed and determine how often $Q_{\max}$ is also observed within these instances. Additionally, we provide general information about the frequency of the different drivers occurring, regardless of whether $Q_{\max}$ has occurred, along with their corresponding discharge values. In this reverse extreme precipitation-based analysis, the highest

discharge is extracted either on the day of the indicator or the next day for $P_{99}$, Compound I, and III, while for other indices, the discharge value on the day of the event is taken to ensure alignment with the extreme discharge event-based analysis.

Antecedent catchment wetness and precipitation depths are crucial factors for floods in terms of magnitudes and volumes. In addition to the aforementioned flood indicators, we delve deeper into this relationship by employing the nonparametric Spearman's rank-order correlation method to correlate the half-year maximum discharge time series ($Q_{\mathrm{W,max}}$ and $Q_{\mathrm{S,max}}$)

with their $k$-day antecedent precipitation depth (denoted as $P_{\mathrm{kD}}$, where $k$ = 1, 3, 5, 7, 10, 15, 30 and 40 days; Tu, 2006). The antecedent precipitation depths are computed by summing the precipitation on the day of the high flow event and the previous $k$ days (Tu, 2006). Table 1 provides an overview of all the different flood indicators used in this study, along with their corresponding definitions.

One crucial concern is the timing difference of the measured daily values, i.e. calendar days for discharges and 24-hour

sums from 8 UTC for precipitation. The reported time/date in the precipitation time series (for the manual rain gauge network; Valkenburg, Ubachsberg, Noorbeek and Vaals stations) is the end time of observation. For this reason, the time of occurrence

**Table 1.** Extreme indicators used to identify the main flood drivers. The initial six indicators are used to calculate probabilities preceding high-flow episodes, whereas the last indicator is correlated with high flows

| Indicator | Estimation method |
|---|---|
| Extreme precipitation ($P_{99}$) | 24-hour precipitation exceeding the 99th percentile of rainy days (more than 1 mm) |
| Multi-day precipitation ($P_{MD}$) | Four-day (Sect. 3.1) accumulated precipitation amount exceeding the 95th percentile thresholds |
| Wet antecedent conditions ($P_{WAC}$) | Antecedent Precipitation Index (API) value higher than 1.5 |
| Compound I | Extreme precipitation occurring on wet antecedent conditions |
| Compound II | Multi-day precipitation occurring on wet antecedent conditions |
| Compound III | Extreme and multi-day precipitation occurring on wet antecedent conditions |
| Antecedent precipitation depths ($P_{kD}$) | Sum of precipitation on the day of the high flow event and the previous $k$ days |

of the max daily discharge is checked: if the time of the maximum 15-min discharge values is observed to be between 00:00 and 8:00 AM the calendar day of this event is reported, otherwise, the next day is considered as the date of occurrence of the event for the (manual rain gauge) precipitation time series. In this way, we ensure the agreement between the precipitation and
discharge time series.

## 2.4    Trend analysis

The temporal variability and the trends of the aforementioned potential flood drivers (i.e. $P_{99}$, $P_{MD}$ and $P_{kD}$) are investigated. Trends in $P_{kD}$ are based on the highest $k$-day total precipitation per year (a summation moving window with different lengths is applied over the whole time series from the 1950s to 2021 and the annual maxima are extracted). In contrast to the definition
of $P_{99}$ in Sect. 2.3.3, we use the 95th percentile, as events that exceed the 99th percentile are extremely rare and result in many zero values in the time series, especially in winter periods, making the trend analysis unstable. The new index is denoted as $P_{95}$ and represents the annual total precipitation from days exceeding the 95th percentile. Similarly to $P_{99}$ the 95th percentile is calculated using the whole range of the time series and only wet days. Finally, the $P_{MD}$ trends are investigated by summing the annual amount of four-day accumulated precipitation that exceeds the 95th percentile thresholds (of the four-day rolling sum).
The used precipitation trend indices are similar to ETCCDI (Expert Team on Climate Change Detection Indices) and have been frequently applied (Klein Tank et al., 2002; Dunn et al., 2020). The aforementioned precipitation indices are calculated for winter and summer periods. In addition, trends in the half-year maximum discharges ($Q_{W,max}$ and $Q_{S,max}$) are examined in order to detect possible consequences of extreme precipitation on extreme streamflows. We use the non-parametric M-K test to detect significant trends. To this end, a Python package that contains all the types/modifications of the M-K test, as developed
by Hussain and Mahmud (2019), is used. The original M-K test is employed on both the precipitation indices and discharge time series, instead of a modified M-K version that accounts for the influence of serial correlation on trend calculations. This choice is guided by the assumption that the precipitation time series exhibit no significant serial correlation and that the annual maximum discharge values are typically considered uncorrelated by construction.

The statistical significance and direction of the trends in the multi-temporal approach are used in order to determine the temporal consistency of a trend for each precipitation index. We express consistency as the percentage of time ($t$) for which trends are statistically significant. Trends that are significant at $\alpha = 0.2$ (Łupikasza et al., 2011) are defined as statistically significant trends. Since precipitation is characterized by strong temporal and spatial variation, the statistical significance levels can be lower compared to other climatic variables (Łupikasza et al., 2011). In addition, the large number of calculated trends (more than 850 for most stations) allows the use of lower significance levels for expressing stabilities or consistencies. In a multi-temporal approach, it is preferable to focus on the direction and intensity of the trends, rather than whether they surpass a strictly predetermined and somewhat arbitrary level of significance (Hannaford et al., 2013). A trend in an index is considered as inconsistent, weakly consistent, consistent and strongly consistent according to the following criteria: (1) inconsistent: 0 % $< t \leq 15$ % or the number of significant increasing and decreasing trends are similar (i.e. the percentages of significant trends in the same direction ranges between 40 % and 60 %); (2) weakly consistent: 15 % $< t \leq 25$ % and more than 60 % of significant trends are in the same direction; (3) consistent: 25 % $< t \leq 45$ % and more than 60 % of significant trends are in the same direction; and (4) strongly consistent: $t > 45$ % and more than 60 % of significant trends are in the same direction.

## 3   Results

### 3.1   Identification of the dominant flood driver

#### 3.1.1   Selection of $P_{\mathrm{MD}}$ duration

In this section, we explore the sensitivity of the probability of $P_{\mathrm{MD}}$ (95th percentile of all $k$ day accumulated time series, see Sect. 2.3.1) occurrences before high flows (i.e. $Q_{\mathrm{max}}$) with respect to the duration of multi-day precipitation. Our analysis reveals that a 4-day duration is the most suitable for defining $P_{\mathrm{MD}}$, as the average relative frequency of $Q_{\mathrm{max}}$ preceded by this duration is the highest compared to other accumulation periods (i.e. $P_{\mathrm{MD}}$ durations of 5, 6, 7, 8, 9, and 10 days; Table 2). This finding aligns with the hydrological behavior of the catchment, as documented by Asselman et al. (2022). Additionally, we observe that the average relative probability of $P_{\mathrm{MD}}$ decreases by approximately 10% when the duration increases from four to five days. However, for durations longer than five days, the frequency of $P_{\mathrm{MD}}$ preceding high flows remains relatively stable. In light of these results, the 4-day duration is utilized as the standard for $P_{\mathrm{MD}}$ definition.

#### 3.1.2   Seasonality of extreme indicators

The seasonal distribution of extreme precipitation, high flow events, and flood drivers indicate that extreme precipitation, although more frequent during summer months, does not consistently coincide with high flow events (Fig. 3). Specifically, Fig. 3 shows the seasonal distribution of extreme precipitation ($P_{99}$; 69 events - about 0.4% of the daily time series from 1970 to 2021), annual maximum precipitation ($P_{\mathrm{max}}$; 52 events), multi-day precipitation ($P_{\mathrm{MD}}$; 933 events - 5% of the four-day accumulated time series from 1970 to 2021) and wet antecedent condition events ($P_{\mathrm{WAC}}$; 2705 events - 14% of the 30-day accumulated time series from 1970 to 2021) at the Vaals station, as an example, as well as high flow events ($Q_{\mathrm{max}}$ and

**Table 2.** Mean relative frequencies for all stations of high flow events preceded by multi-day precipitation ($P_{MD}$) for different accumulation periods.

| $P_{MD}$ duration | $Pr(P_{MD} \mid Q_{max})$ [%] |
| --- | --- |
| 4-day | 74.7 |
| 5-day | 65.3 |
| 6-day | 64.5 |
| 7-day | 61.6 |
| 8-day | 57.9 |
| 9-day | 61.2 |
| 10-day | 58.4 |

$Q_{99}$; 49 and 91 events, respectively) together with past flood events (30 events) in the catchment. General information on the frequency of the flood drivers averaged for all rainfall stations, is provided in Table 5. An opposing seasonality is visible between extreme precipitation and high flow events. The relative frequencies of $P_{99}$ and $P_{max}$ in half-year summer periods are 75% and 73%, respectively, while these percentages are only 22% and 29%, for the $Q_{max}$ and $Q_{99}$ events. This pattern is also verified by the past flood events: only 26% of them occurred during half-year summers. $P_{MD}$ appears to occur with relatively

similar frequencies throughout the year. Similar to $P_{MD}$, $P_{WAC}$ shows an equal distribution all over seasons as expected. Since precipitation is rather uniformly distributed all over the year (see Fig. 2a) wetter than average conditions can occur in any season, and monthly distribution of $P_{WAC}$ cannot be directly linked to the seasonal distribution of high flows. However, wetter conditions (indicated by higher API values) are expected to have different effects on high flows, and a closer examination of the specific condition before every high flow event is necessary. Overall, our findings indicate that extreme 24-hour precipitation

is not the most critical driver of high flows. Although extreme precipitation events tend to occur more frequently during the summer months, high flow and flood events do not align with these periods. Factors such as antecedent soil moisture conditions, as well as the timing, duration, and intensity of rainfall events, may exert a more significant influence on high-flow generation in the catchment. Therefore, greater attention is required in understanding these factors.

### 3.1.3    Extreme discharge event-based analysis

Figure 4 illustrates the relative frequencies of the introduced indicators (see Table 1) preceding high-flow episodes (i.e., $Q_{max}$), as described in Sect. 2.3.3. In approximately 75% of the $Q_{max}$ cases, a $P_{MD}$ precedes high-flow events (Fig. 4b), while the corresponding percentage for $P_{WAC}$ is approximately 48% (Fig. 4c). In most cases, $P_{WAC}$ should be followed by $P_{MD}$ in order to generate high flows (similar percentages between $P_{WAC}$ (Fig. 4c) and Compound II (Fig. 4e)). This is also visible in the calculation of the conditional probability of $P_{MD}$ preceding $Q_{max}$ given that $P_{WAC}$ precedes $Q_{max}$ which is approximately

84% (Fig. 4e). In other words, given that $P_{WAC}$ precedes $Q_{max}$, there is 84% chance that $P_{MD}$ will be followed by $Q_{max}$. The percentages between Compound I and Compound III events (Fig. 4d and 4f) are the same, showing that each time a $Q_{max}$ is

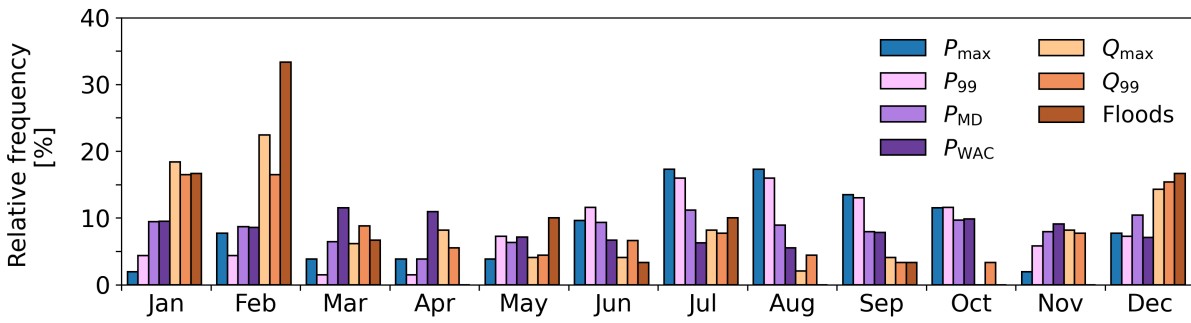

**Figure 3.** Monthly distribution of annual maximum precipitation ($P_{\mathrm{max}}$), extreme precipitation ($P_{99}$), multi-day precipitation ($P_{\mathrm{MD}}$) and wet antecedent condition events ($P_{\mathrm{WAC}}$) at the Vaals station, annual maximum discharge ($Q_{\mathrm{max}}$), extremely high flows ($Q_{99}$) and past flood events in the Geul.

preceded by $P_{99}$ it also preceded by $P_{\mathrm{MD}}$. This effect in most cases is caused by the high amount of precipitation that fell on the day of the event or the previous day (definition of $P_{99}$) which also increases the 4-day precipitation (higher than the $P_{\mathrm{MD}}$ 95th percentile).

Figure 5 shows the $Q_{\mathrm{max}}$ events plotted against their API, including also their preceding precipitation indicators (i.e. $P_{99}$ and $P_{\mathrm{MD}}$) at the Maastricht and Vaals stations. This figure actually presents how the different events are classified based on the preceded defined indicators (Table 1), emphasizing the influence of wet conditions on high flows and exploring correlations between $Q_{\mathrm{max}}$ and associated precipitation amounts ($P_{99}$ or $P_{\mathrm{MD}}$). For example, all $P_{\mathrm{MD}}$ markers (both orange and purple markers), irrespective of their wetness (API), are classified as $P_{\mathrm{MD}}$ and thus used to calculate the relative frequencies of $Q_{\mathrm{max}}$

being preceded by $P_{\mathrm{MD}}$ in Fig.4b. Furthermore, Fig. 5 reveals overlapping event classifications, where one event can align with multiple indicators at the same time, e.g. a Compound III event is classified as $P_{\mathrm{WAC}}$, $P_{99}$, $P_{\mathrm{MD}}$, Compound I and II, while a $P_{99}$ event can be at the same time $P_{\mathrm{MD}}$. A $Q_{\mathrm{max}}$ event preceded by $P_{99}$ may appear in the "Wet" classification without being classified as Compound I. This is because we require that $P_{99}$ should occur under existing $P_{\mathrm{WAC}}$ conditions to be classified as Compound I. This condition is imposed to prevent a $P_{99}$ event from inflating the API the day before the event, potentially

leading to an API > 1.5 on the day of the event (see Sect. 2.3). For example, the 1970 event at Vaals (Fig. 5d), which is preceded by $P_{99}$, $P_{\mathrm{MD}}$, $P_{\mathrm{WAC}}$, and Compound II, is not classified as Compound I or III.

    All high flow events appear to happen during normal or wet conditions (Fig. 5). Most extreme events are caused by a combination of $P_{99}$ and $P_{\mathrm{MD}}$ or just $P_{\mathrm{MD}}$. Antecedent conditions play also a crucial role in translating precipitation extremes to high flows. We observe that higher API values (higher wet initial conditions) lead to higher peak discharges, especially in

events preceded by compound and $P_{99}$, where a strong correlation between $Q_{\mathrm{max}}$ and API is observed while a very weak correlation between $Q_{\mathrm{max}}$ and total event $P_{\mathrm{MD}}$ or $P_{99}$ precipitation is reported (see Fig. 5). $P_{\mathrm{WAC}}$ makes the difference between a "regular high flow" and a flood event. This is also evident in the initial conditions of the top five floods that occurred in the catchment during the study period (Fig. 5 and Table 3). Details about these events (i.e date of occurrence, different rainfall accumulations, and an estimate of their initial conditions) are presented in Table 3 based on the precipitation records

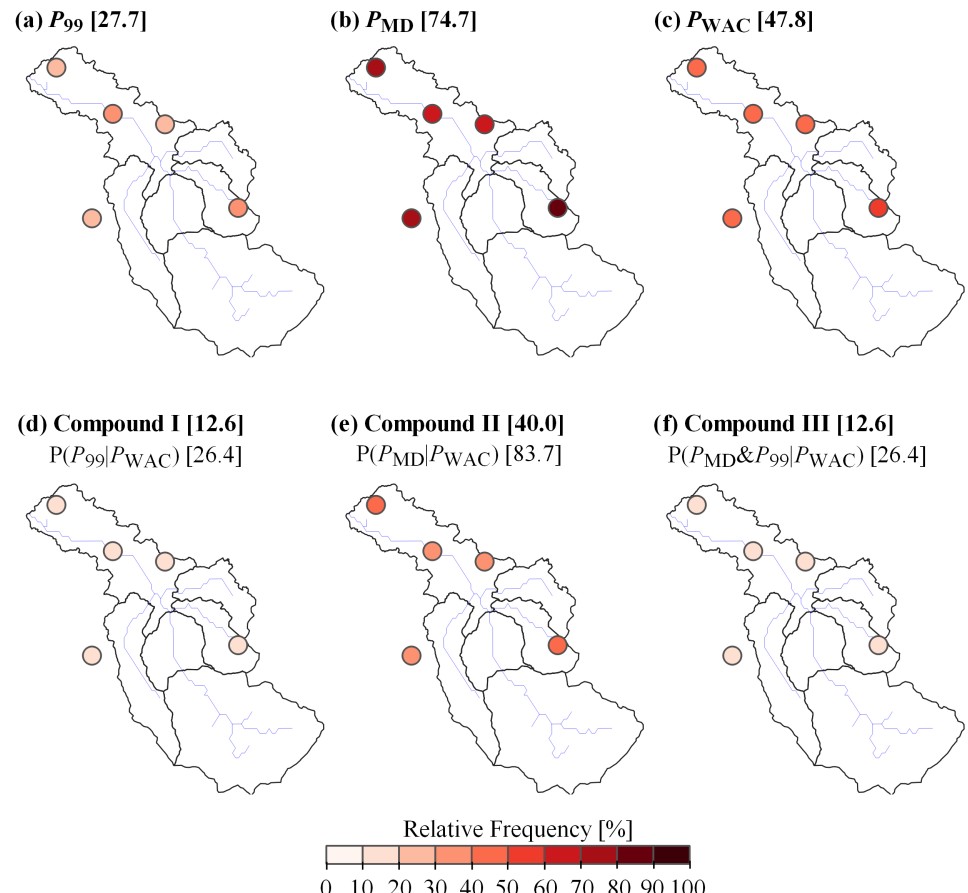

**Figure 4.** Relative frequencies of $Q_{max}$ being preceded by (a) extreme precipitation ($P_{99}$), (b) multi-day precipitation ($P_{MD}$), (c) wet antecedent conditions ($P_{WAC}$), (d) extreme precipitation on wet antecedent conditions (Compound I), (e) multi-day precipitation on wet antecedent conditions (Compound II), and (f) extreme and multi-day precipitation on wet antecedent conditions (Compound III), in the total $Q_{max}$ events (count of a driver leading to $Q_{max}$ in the $Q_{max}$ cases divided by the total number of cases). The mean relative frequency (in %) for all rainfall stations is reported in brackets. The mean conditional probabilities of $P_{99}$ and/or $P_{MD}$ preceding $Q_{max}$ given that $P_{WAC}$ precedes $Q_{max}$ are also reported per Compound indicator.

at Maastricht and Vaals. Examining the preceding conditions for the major past floods, it appears that in most of these cases, while the precipitation events spanning 1 to 3 days were heavy, the overall precipitation over the 30 days preceding the events was substantial. This extended period of precipitation likely played a critical role in saturating the catchment, making it more susceptible to flooding. The combination of intense rainfall over shorter durations and continuous precipitation over the 30-day period seemed to collectively contribute to the formation of wet initial conditions, ultimately increasing the risk and eventually

resulting in flooding. It is also important to note that the large discrepancies in rainfall accumulations between the two stations

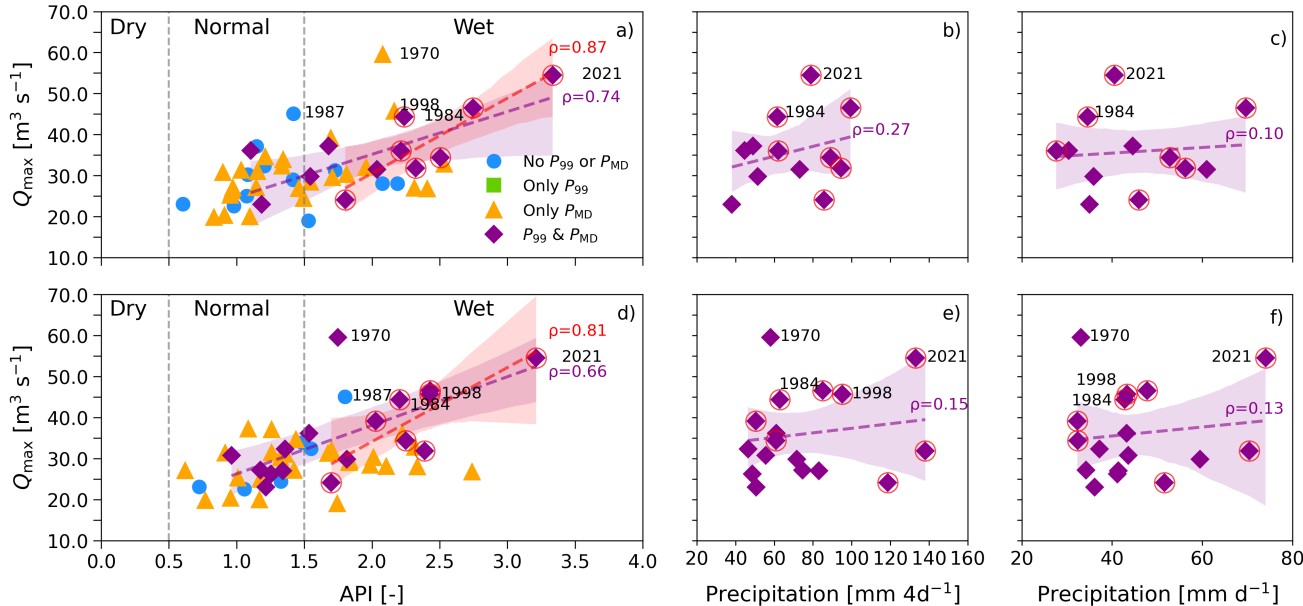

**Figure 5.** Annual maxima events ($Q_{max}$) and their Antecedent Precipitation Indices (API) at Maastricht (a) and Vaals (d), including their preceding extreme indicators. Orange markers denote events preceded solely by $P_{MD}$, green markers indicate events preceded exclusively by $P_{99}$, purple markers represent events preceded by both $P_{99}$ and $P_{MD}$ (thus classified as both $P_{99}$ and $P_{MD}$ in Fig. 4), and blue markers signify events without any extreme precipitation indicator preceding them. Purple and orange markers within the "Wet" classification, along with $P_{WAC}$, $P_{99}$, and/or $P_{MD}$ classifications, are also classified as Compound II events in Fig.4. Discharge events preceded by Compound III (and thus Compound I and II) are indicated with red circles. The top five floods during the study period are shown with their year of occurrence. The dashed purple line represents the linear fit, using the least squares approach, between the API of the high flow events preceded by $P_{99}$ and $P_{MD}$ and their respective $Q_{max}$ values, while the red dashed line represents the linear fit between the API of the Compound III events and their $Q_{max}$. The total four-day precipitation versus $Q_{max}$ is presented for these events at Maastricht (b) and Vaals (e), and also the highest 24-hour precipitation (highest of the two $P_{99}$ amounts on the day of the event or the previous day) versus $Q_{max}$ at Maastricht (c) and Vaals (f). The shaded area shows the 95% confidence intervals for the fits, and the Pearson's correlation coefficients ($\rho$) are also reported.

found in Table 3 are caused mainly by the different time interval they include, i.e calendar days for the Maastricht station and 24-hour sums from 8 UTC for the Vaals station, but also the spatial variability.

The absence of green markers in Fig. 5 (only $P_{99}$ events) indicates again that for days preceding $Q_{max}$ the probability of a $P_{MD}$ event given that $P_{99}$ occurs equals one (Pr ( $P_{MD}$ | ($P_{99}$ & $Q_{max}$) ) = 1). In the way we defined $P_{99}$ (two days interval) we observe that the $P_{99}$ events preceding a $Q_{max}$ usually coincide with also $P_{MD}$. For very extreme 24-hour events the 4-day 95th percentile used for the $P_{MD}$ definition can be exceeded and cause at the same time both $P_{MD}$ and $P_{99}$, which is unavoidable. However, in longer accumulation periods for $P_{MD}$ (i.e. 5, 6, 7, 8, 9, and 10 days) the corresponding 95th percentile increases, as the moving/accumulated period is extended, and becomes much larger than the 99th percentile used for the definition of $P_{99}$. In these cases, irrespective of the duration the mean relative frequencies of high flows preceded by Compound II and III

**Table 3.** Top five floods that occurred in the catchment during the study period, their rainfall accumulations, and an estimate of their initial conditions based on API. The day with the highest recorded discharge is considered as the date of occurrence of the event. The different precipitation sums (and subsequently API) are based on daily precipitation records at the Maastricht (first value) and Vaals (second value) stations.

| No. | Date | 24-hour sum [mm] | 48-hour sum [mm] | 72-hour sum [mm] | 30-day before event sum [mm] | API [-] | Initial condition |
|---|---|---|---|---|---|---|---|
| 1 | 15/07/2021 | 13.1 / 6.9 | 53.6 / 80.9 | 78.0 / 131.5 | 244.7 / 274.5 | 3.33 / 3.21 | Wet / Wet |
| 2 | 15/09/1998 | 9.2 / 28.1 | 31 / 71.4 | 48.7 / 74.7 | 135.5 / 187.2 | 2.16 / 2.42 | Wet / Wet |
| 3 | 28/02/1987 | 0.0 / 0.0 | 13.1 / 13.9 | 14.5 / 23.1 | 83.1 / 141.8 | 1.42 / 1.80 | Normal / Wet |
| 4 | 07/02/1984 | 22.5 / 10.4 | 57.1 / 53.2 | 60.5 / 58.3 | 131.4 / 170.3 | 2.24 / 2.20 | Wet / Wet |
| 5 | 22/02/1970 | 15.0 / 14.1 | 35.0 / 47.2 | 38.0 / 53.4 | 125.4 / 140.5 | 2.08 / 1.75 | Wet / Wet |

remain stable (see supplementary material for the analysis). This implies that preceding $P_{99}$, rainfall events (whether heavy or not) probably occurred for these events as well (at least for less extreme ones), potentially resulting in wet conditions and consequently high discharges, highlighting the correlation among the used different drivers and how they can be converted to compounds. Thus, while it is found that $Q_{max}$ is preceded by $P_{MD}$ 75% of the time, some of the $P_{MD}$ events could be forced or even caused by $P_{99}$. However, the definition of $P_{MD}$ still holds significance as it denotes an extended period of heavy rainfall.

Finally, we also explore the effect of antecedent precipitation depths on high flows, as an indicator of antecedent catchment wetness. Table 4 reports the correlation coefficients of the $Q_{W,max}$ time series with their $k$-day ($k$ = 1, 3, 5, 7, 10, 15, 30 and 40) antecedent depths per rainfall station. Results are presented only for the winter half-year as approximately 80% of $Q_{max}$ occurred in this period (Fig. 3). Thus, $Q_{S,max}$ time series contains mainly low discharge values (most of them lower than 20 m$^3$ s$^{-1}$, see Fig. 10a) with meaningless correlations. Peak half-year winter discharges in the Geul appear to be closely related

to antecedent 10-40 days precipitation depths on wet soils (e.g. correlation coefficient of 0.66 in Vaals for the duration of 15 days). The effect of wet antecedent conditions is also reflected here. These results seem reasonable considering the opposing seasonality observed in the catchment (Fig. 3). The July 2021 flood event is consistent with these results as there is evidence that the wetness of the catchments was much higher than usual for the time of the year (see Fig. 5 and Table 3).

### 3.1.4 Extreme precipitation based analysis

In this section we approach the problem from the reverse way compared to Sect. 3.1.3. Table 5 shows the frequency of the potential flood drivers actually leading to extreme discharge. $P_{99}$, Compound I, and Compound III are rare events compared to the other drivers, and there are years in which they are not observed at all (see Table 5). In the unique years when a $P_{99}$ event occurs, it is followed by a $Q_{max}$ event in about 37% of those cases, while these numbers are approximately 65% for Compound I and Compound III, respectively (Table 5). Compound I and III events are extremely rare events (approximately

13 recorded events per station), however in most cases when they appear they lead to $Q_{max}$. This observation suggests that

**Table 4.** Correlation coefficients between the winter half-year discharge maxima and their antecedent $k$-day precipitation depths.

| Station | $P_{\mathrm{kD}}$ - Winter half-year | | | | | | | |
|---|---|---|---|---|---|---|---|---|
| | $k{=}1$ | $k{=}3$ | $k{=}5$ | $k{=}7$ | $k{=}10$ | $k{=}15$ | $k{=}30$ | $k{=}40$ |
| Vaals | 0.44 | 0.32 | 0.44 | 0.40 | 0.59 | 0.66 | 0.56 | 0.50 |
| Valkenburg | 0.30 | 0.20 | 0.26 | 0.28 | 0.47 | 0.57 | 0.52 | 0.44 |
| Ubachsberg | 0.26 | 0.21 | 0.25 | 0.30 | 0.46 | 0.56 | 0.53 | 0.46 |
| Noorbeek | 0.36 | 0.27 | 0.36 | 0.34 | 0.54 | 0.61 | 0.55 | 0.50 |
| Maastricht | 0.24 | 0.15 | 0.27 | 0.27 | 0.41 | 0.53 | 0.49 | 0.43 |
| **Average** | **0.32** | **0.23** | **0.31** | **0.31** | **0.49** | **0.59** | **0.53** | **0.47** |

extreme precipitation $P_{99}$ when occurs under wet antecedent conditions, leads to high flows, but alone is typically insufficient to cause floods.

**Table 5.** General information, averaged for all rainfall stations, regarding the frequency of the different drivers occurring regardless of $Q_{\mathrm{max}}$ having occurred, in the $Q_{\mathrm{max}}$ period (hydrological years from 1970 to 2021, excluding 1971, 1974 and 1990).

| | Number of events | Number of unique years occurred | Mean yearly occurrence | Number of years followed by $Q_{\mathrm{max}}$ | Relative (reverse) frequency |
|---|---|---|---|---|---|
| | (1) | (2) | (1) / (2) | (3) | (3) / (2) |
| $P_{99}$ | 64.2 | 36.4 | 1.77 | 13.6 | 0.37 |
| $P_{\mathrm{MD}}$ | 873.6 | 49.0 | 17.8 | 36.6 | 0.75 |
| $P_{\mathrm{WAC}}$ | 2760.6 | 48.4 | 57.0 | 23.6 | 0.49 |
| Compound I | 13.4 | 10.0 | 1.34 | 6.2 | 0.62 |
| Compound II | 319.6 | 43.6 | 7.33 | 19.8 | 0.45 |
| Compound III | 13.0 | 9.6 | 1.35 | 6.2 | 0.65 |

It appears that flooding is rarely caused by a single mechanism. Figure 6 shows the discharge empirical cumulative distribution functions for every station caused by every extreme precipitation indicator, for the number of events reported in Table 5. According to the local water authorities, a discharge exceeding approximately 40 $\mathrm{m}^3\mathrm{s}^{-1}$ could lead to flooding of the first houses, while a discharge greater than 30 $\mathrm{m}^3\mathrm{s}^{-1}$ could result in the inundation of floodplains, particularly concerning upstream locations within the catchment area (Klein et al., 2023). These thresholds are often crossed during Compound I and III events, particularly in extreme events at the Vaals station, indicating a stronger correlation with discharges compared to other locations. It is important to note that these discharge thresholds pertain to the upstream sections of the catchment and not the outlet at Meerssen. However, most of the water originates from the Belgian part while the contribution of the remaining tributaries is minor (Klein, 2022). Overall, it seems that single processes are not likely to cause floods, whereas compound events do. It is also again clear that $P_{99}$ in isolation is much less likely to cause floods compared to when combined with wet antecedent conditions and $P_{\mathrm{MD}}$.

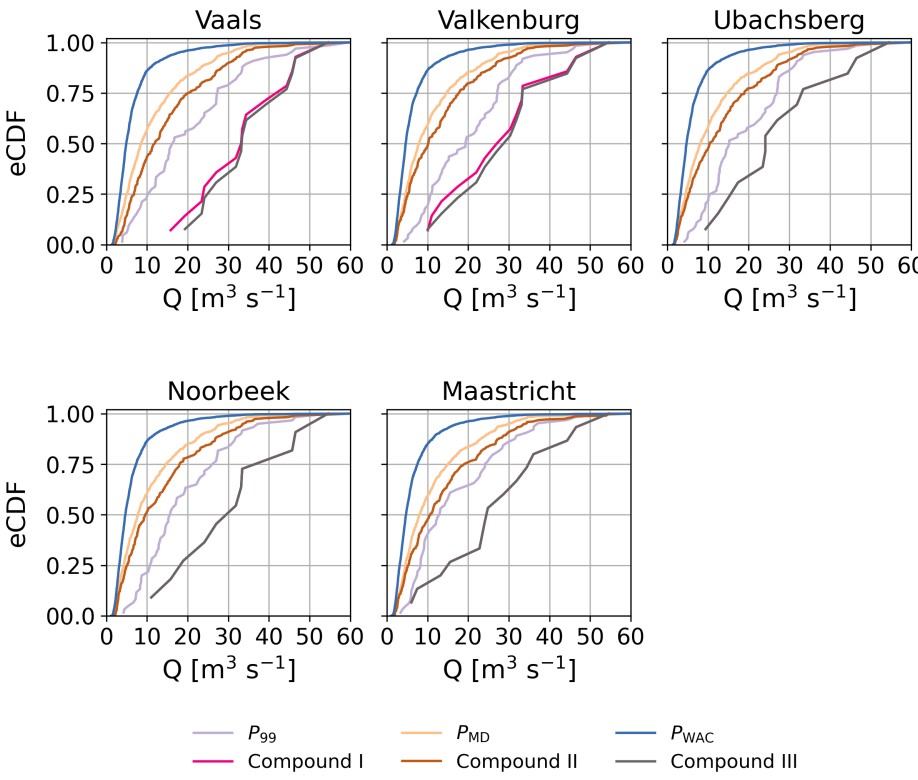

**Figure 6.** Empirical Cumulative Distribution Functions (eCDFs) of discharge caused by the extreme precipitation indicators for all stations, based on the number of events from Table 5

### 3.1.5 Sensitivity of API to evaporation

The exclusion of evaporative processes in the API used for evaluating initial catchment conditions could potentially pose significant concerns, as the index is solely based on antecedent precipitation depths. An important aspect to consider is whether API reliably reflects soil wetness consistently throughout the year in our study region or if its interpretation is influenced by the strong seasonal variations (reference evaporation is low in winter and high in summer, see Fig. 2). To investigate this, the simple 30-day before an event effective rainfall (precipitation minus reference evaporation from the Maastricht station) is

calculated instead of the simple API, and Fig. 5 is reproduced.

The results suggest that no significant biases are occurred due to the exclusion of evaporation. Larger offsets are visible in the summer half-year events and the overall correlation is low (as in the regular API, Fig. 5), however, the high correlation between the $P_{99}$ and $P_{MD}$ events and the $Q_{max}$ (purple markers and reported correlation value) is maintained (Fig. 7), which is in line with the API results. Only 7/49 $Q_{max}$ events occurred in summer and 6/7 summer $Q_{max}$ events are in the $P_{99}$ and

$P_{MD}$ events and as a result the calculated correlation includes them. In addition, the top five floods remain "higher" compared to other events.

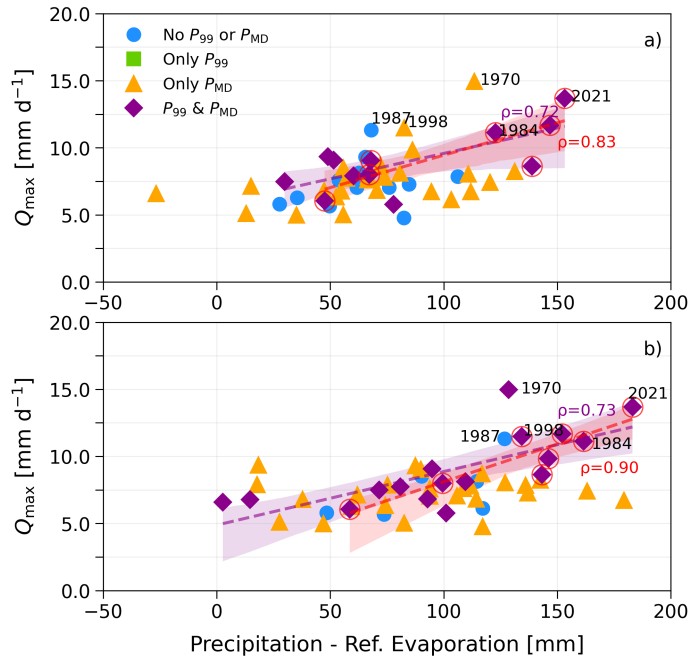

**Figure 7.** Annual maxima events ($Q_{\mathrm{max}}$) and their 30-day pre-event effective rainfall at Maastricht (a) and Vaals (b), including their preceding extreme indicators. The top five floods during the study period are shown with their year of occurrence. The dashed purple line represents the linear fit, using the least squares approach, between the effective rainfall of the high flow events preceded by $P_{99}$ and $P_{\mathrm{MD}}$ and their respective $Q_{\mathrm{max}}$ values, while the red dashed line represents the linear fit between the effective rainfall of the Compound III events and their $Q_{\mathrm{max}}$. The shaded areas show the 95% confidence intervals for the fits and the Pearson's correlation coefficients ($\rho$) are also reported.

## 3.2 Trend analysis

### 3.2.1 Flood driver trends

Figure 8 illustrates the multi-temporal trend analysis for several precipitation indices for the Vaals station, for half-year periods,

as an example. The multi-temporal analysis for Vaals, which has a record from 1952 to 2021, results in 861 trends. In the winter half-year statistically significant increasing trends are found for the longest periods in all indices at Vaals. However, a decreasing, mainly insignificant, pattern is visible in the recent past (trends starting after the 1980s). In summer (Fig. 8b) negative trends are visible for the longest periods, while this changes to positive trends in the recent past for $k \leq 5$ days. Summer trends for $k > 5$ days are rather mixed: generally insignificant trends, with shifts between positive and negative tendencies. For

the full multi-temporal analysis per index and station, please refer to the supplementary material.

Figure 9 shows the consistency of statistically significant trends in each precipitation index per rainfall station. In winter only increasing trends are visible, with the exception of the $P_{15\mathrm{D}}$ index at Ubachsberg. The decreasing tendency in the recent past detected at Vaals is very strong and statistically significant at Ubaschsberg for most indices (Fig. S3 in supplement), which

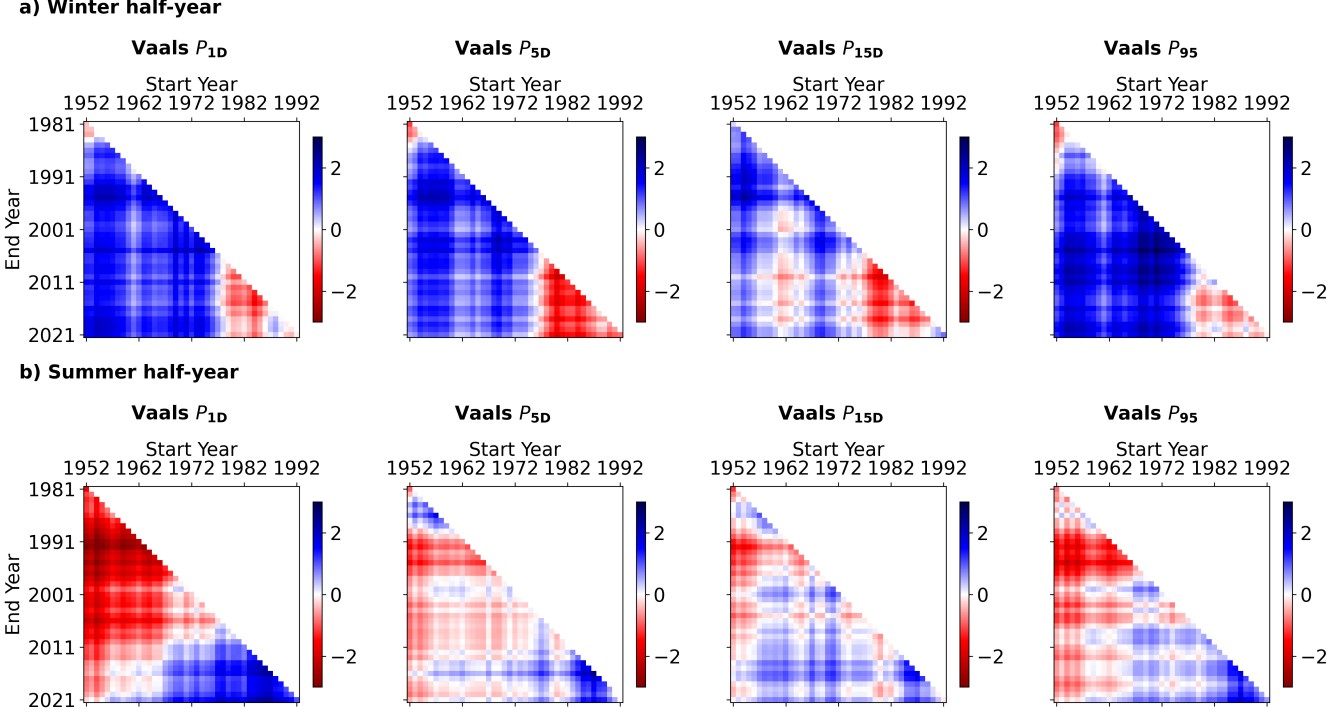

**Figure 8.** Multi-temporal trend analysis for $P_{1D}$, $P_{5D}$, $P_{15D}$ and $P_{95}$ at Vaals for (a) winter half-year and (b) summer half-year. Each pixel presents a fixed single period (minimum window length of 30 years) of start and end year. For each period the M-K test is applied, and the color indicates the Z-statistic value of the test (the same definitions apply to the subsequent figures). Blue colors indicate increasing trends and red downward ones. The darker the color, the more significant the trend. Statistically significant trends are considered those with Z-statistic values higher than 1.28 (or smaller than -1.28 for downward trends) corresponding to the defined significance level of 0.2 (see Sect. 2.4).

causes this inconsistent decrease. Consistent and strongly consistent increases are observed in at least one station per index. In the index $P_{7D}$ four out of five stations show consistent or strongly consistent increases. For $k \leq 10$ days most stations have strongly consistent or consistent upward directions. With increasing $k$ (15,..,40 days) the increase becomes weaker (inconsistent) for the majority of the stations, however still two out of five stations (located inside the catchment) show increases. Indices for $k \leq 10$ days are strongly consistently increasing, while for $k > 10$ days a consistent increase is visible in the station Valkenburg. In addition, three out of five stations show strongly consistent or consistent increasing trends in the indices $P_{95}$ and $P_{MD}$. Overall, the trend analysis in winter for the Geul catchment shows a consistent increase of very wet days ($P_{95}$) and maximum $k$-day precipitation sums. The rise in severe precipitation is caused mostly by more rain on already wet days. Multi-day precipitation extremes are consistently increasing. This is a crucial finding, as the effects of prolonged heavy storms in combination with wet antecedent conditions appear to be the dominant flood drivers in the Geul catchment.

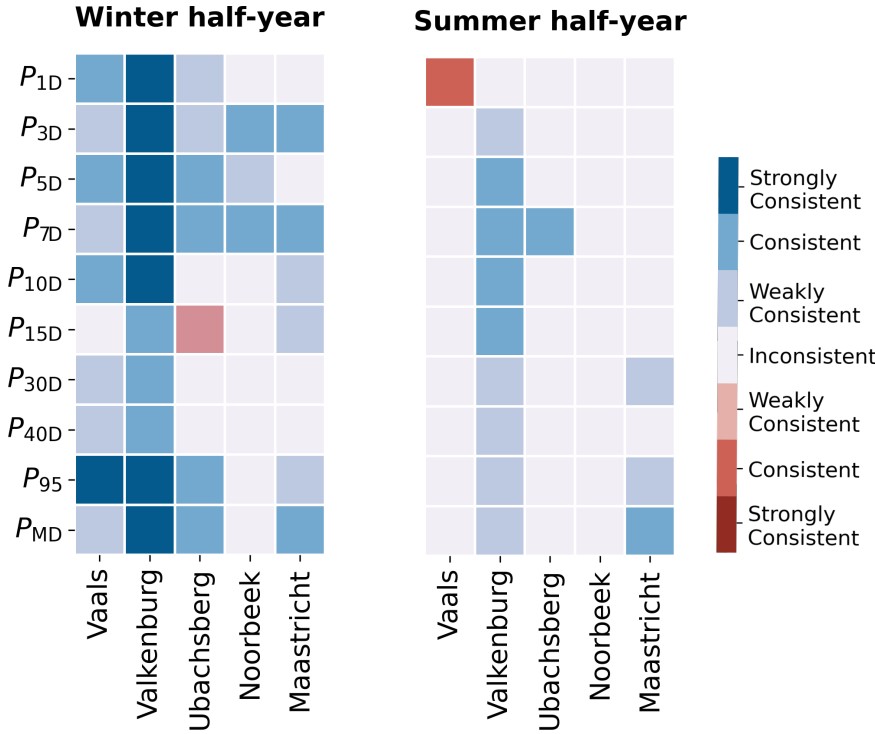

**Figure 9.** Temporal consistency of precipitation trend indices for winter and summer half-years. Blue colors indicate upward trends while red colors indicate downward trends.

Trends in summer periods show variability across the different stations. The majority of the summer half-year $k$-day and $P_{95}$ indices are subject to inconsistent trends. Most stations show generally insignificant trends, with changes between positive and negative tendencies. The only consistent trends are mainly increasing and are found at the Valkenburg station for $k = 5, 7,$ 10, 15 days, at Ubachsberg for $k = 7$ days, and at Maastricht (i.e $P_{\mathrm{MD}}$). In addition, the $P_{30D}$ and $P_{95}$ indices at Maastricht show strong and statistically significant increasing trends in the majority of tested cases resulting in weakly consistent trends. As mentioned, despite the fact that the difference between significant increasing and decreasing trends in the summer half-year is not clear, the statistically significant increasing trends in the recent past, mainly for $k \leq 3$ days, are strong and should be taken into account (Fig. S6 - S10 in supplement). In addition, the consistent increasing trends at Valkenburg reveal a direction towards more wet conditions in the summer half-year.

### 3.2.2 Discharge trends

The results of the multi-temporal analysis for the $Q_{\mathrm{W,max}}$ and $Q_{\mathrm{S,max}}$ time series are shown in Fig. 10. It can be observed that the maximum flows show variability over the two half-year periods. Increasing trends are found in the longest periods for the winter half-year but this seems to have changed in the recent past to statistically insignificant decreasing tendencies.

Overall, the increase in the $Q_{\mathrm{W,max}}$ is considered consistent taking also into account the missing hydrological years of 1971, 1974 and 1990. This pattern is in agreement with the extreme precipitation trends in the area for winter, as large similarities are observed in terms of magnitudes, directions and variabilities. For example, $Q_{\mathrm{W,max}}$ trends variability is quite similar to

425 $P_{95}$ index in winter at Vaals (see Fig. 8a): statistically stronger increasing trends in longer periods (from 1970 to 2021) with a weak decreasing direction in the recent past (trends starting from the 1980s).

Mixed and non statistically significant trends are observed in the summer half-year (Fig. 10c), as expected, considering that the trends in extreme precipitation in the same period are inconsistent and their strength is (statistically) insignificant. $Q_{\mathrm{S,max}}$ trends shift between negative and positive tendencies in similar, for some cases, periods with the extreme summer precipitation,

however this match is not so clear as in the winter period. The increasing direction of extreme precipitation in the recent past for summer starts becoming visible also in the $Q_{\mathrm{S,max}}$ direction (see positive values for trends starting after 1985). In general, in summer the effect of the considerable increase in evaporation in the area (Tsiokanos, 2022) in combination with the large soil moisture deficits should be taken into account when translating extreme precipitation to extreme streamflows, and subsequently discussing correspondences and differences between discharge and precipitation trends.

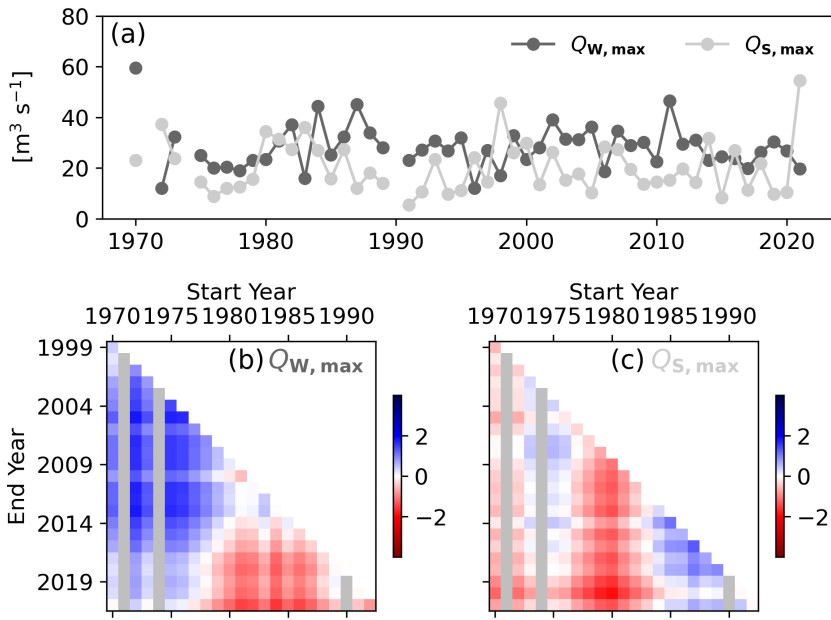

**Figure 10.** Half-year yearly maxima time series (a) and multi-temporal trend analysis for (b) winter half-year yearly maxima ($Q_{\mathrm{W,max}}$) and (c) summer half-year yearly maxima ($Q_{\mathrm{S,max}}$). The gray stripes in the heatmaps indicate the excluded hydrological years (i.e. 1971, 1974 and 1990).

# 4 Discussion

## 4.1 Data uncertainty

Records of 24-hour precipitation are used that come from the KNMI manual rain gauge network. The used precipitation time resolution may be considered low for flood analysis, however the response time (i.e. longer than a day, see Sect. 2.1) of the catchment allows the application of this resolution. In addition, the main goal of this paper is to investigate the role of $P_{99}$, $P_{MD}$, and $P_{WAC}$ as potential flood drivers. The use of 24-hour resolution can affect the defined $P_{99}$, while the applied resolution does not have a major impact on the definitions and meaning of $P_{MD}$ and $P_{WAC}$. For this reason, allowances were made in the way we define a $P_{99}$ day (i.e. precipitation amount at the same or previous day of the high flow event; see Sect. 2.3.3).

Long precipitation time series may have been influenced by instrumental modifications and station relocations throughout the recording period. As mentioned in Sect. 2.2, the data are considered to be of high quality, as KNMI performs regular quality tests. In addition to that, two homogeneity tests were applied to monthly sums. In general, it is assumed that the analyzed precipitation time series in this research are not affected by instrumental and location alterations, so the trends that are found can be attributed to climate and not to human interventions.

It must be acknowledged that the produced discharge results are subject to significant uncertainty (Di Baldassarre and Montanari, 2009). Estimations during extremely high flows are very inaccurate. For example, the recorder discharge in July 2021 flood was 55 m$^3$s$^{-1}$, while it is estimated that it exceeded 80 m$^3$s$^{-1}$ (van Heeringen et al., 2022). However, our main findings about the role of compound events in generating high flows remain valid. In addition, periods of transition, changes in gauge position, equipment, and monitoring frequencies, and stage-discharge relations can cause sudden variations in flow rates. These changes can be more visible in mean flow trends where the values are low in contrast to high flows used in this paper. The long-term measurements of the station Meerssen at the outlet of the catchment are considered reliable in terms of homogeneity (Agor, 2003). The high flow trends are found to be similar to the directions and significance of the extreme precipitation trends, indicating that the results are likely not affected.

## 4.2 Implications

It is found for the Geul that extreme daily precipitation is not solely a flood driver. Wet antecedent soil conditions are a crucial factor determining the probability of flooding. In this respect, the finding that heavy prolonged precipitation frequently preceded high flows in the Geul seems reasonable, as multi-day precipitation can also serve as a proxy for heavy precipitation occurring in wet antecedent circumstances (Nanditha and Mishra, 2022). Most of the flood events are observed in winter periods, when the catchment tends to be very wet, with shallow groundwater tables. In summer periods most of the extreme (intense) precipitation events are not translated to high flow peaks, due to large soil moisture deficits. The most devastating flood in the area (i.e. July 2021) was aggravated by rainfall events in preceding days and weeks. The role of wet antecedent (soil) conditions in driving floods is well established, however, the focus tends to be on larger catchments (Wasko et al., 2020). Information about the initial (wet) conditions of the catchment is deemed essential, particularly for flood forecasting, since the local water authority currently does not monitor soil moisture. In addition, the geology of the Geul can significantly control

the runoff response, as there is a thick unsaturated chalk zone that can store much water (Klein, 2022). The (geo)hydrological properties and characteristics of the catchments should receive more attention in the flood forecasting system (Zanon et al., 2010; Douinot et al., 2022). Our findings are also expected to help in the understanding of flood mechanisms in other lowland or chalk catchments around the world. In addition, the followed event-based approach can be exploited in other catchments to examine the relative role of wet antecedent soil moisture conditions and precipitation characteristics preceding high flows, especially in areas where (long-term) soil moisture data are not available.

The statistical results obtained in Sect. 3.2.1 demonstrate some intriguing variations in the Geul catchment's precipitation regimes across the studied periods. The most notable change is the consistent and strong increase in critical precipitation during the winter half-year. During this period, various indices representing heavy prolonged events such as $P_{kD}$ for $k \geq 3$ days and/or $P_{MD}$, as well as 24-hour extreme indices like $P_{1D}$ and $P_{95}$, show mostly consistent increases. These combinations of indicators can contribute to the saturation of the catchment, thereby increasing the risk of flooding. In addition, it appears that a portion of the rise in severe precipitation stems from increased rainfall on already wet days, as evidenced by consistent (or strongly consistent) rises in $P_{3D}$, $P_{5D}$, $P_{7D}$, and $P_{MD}$ across the majority of stations. All these findings are crucial as heavy and prolonged storms in combination with wet antecedent conditions have impacted the catchment and caused floods mainly in winter. Although it cannot be concluded that climate change had a significant impact on the July 2021 flood event in the Geul region, as there are no apparent consistent patterns in most summer precipitation trends, a concerning increasing direction in the recent past (mainly after the 1980s) is visible. This finding is important as it shows that, except for the intense showers in summer, the effects of heavy storms in combination with wet antecedent conditions should be also taken into account. At the same time, it is critical to consider the substantial increase in summer potential evaporation rates (due to increases in temperature and radiation) in the area (Tsiokanos, 2022) that may lead to soil moisture deficits, when translating extreme precipitation events into potential extreme flows. According to the recently published KNMI scenarios wetter winters and increased extreme summer showers are projected for the Netherlands (KNMI, 2023). These projections suggest that the number of heavy showers with significant precipitation is expected to rise, with a shift from light to heavier (more rain falls from the shower) and more intense (more rain falls in a certain time) showers (KNMI, 2023). These climate scenarios are in line with the trends found in this study. Overall, the long-term trends of the critical precipitation are also visible in the runoff patterns. Thus, climate change should be taken seriously into account in the area and should be incorporated into flood designs, considering also the effects of agro-economic developments, such as land-use changes. Our findings from the precipitation and discharge trend analyses can serve as a valuable reference for assessing the impact of climate change on precipitation and discharge patterns in other regions than the Geul as well. In addition, the use of a multi-temporal approach, including the consistency criteria, appears useful and is recommended for identifying variability, recent directions but also long-term trends.

## 5  Conclusions

We used an event-based approach to detect the main flood drivers in the Geul river catchment and a multi-temporal trend analysis to investigate their temporal variability and consistency. Our results indicate that heavy multi-day precipitation can

have a notable impact on high flows, preceding them approximately 75% of the time (using a 4-day precipitation duration; Fig. 4). Nevertheless, wet antecedent conditions play a crucial role in translating extreme precipitation events into extreme flows and make the difference between a "regular high flow" and a flood event. Extreme 24-hour precipitation, without wet antecedent conditions, which appears mainly in summer, has never led solely to floods in the past (Fig. 5 and Table 3). The joint probability of extreme (prolonged) rainfall and wet initial conditions (which can be seen as a compound event) determines the chances of flooding. As a result, prolonged heavy rainfall and wet antecedent wetness appear to be the primary factors contributing to extreme discharge events, and they should be used as flood indicators, rather than extreme precipitation alone. Flood producing precipitation shows a consistent increase in the winter half-year, a period in which more than 70% of extremely high flows have occurred historically. Heavy prolonged storms in combination with wet antecedent conditions can cause large flooding and these conditions are becoming more frequent during winters. This rise is also reflected in the winter half-year maximum discharges, which are increasing in terms of magnitude. Although the majority of precipitation and flow trends do not exhibit consistent patterns in the summer half-year, a notable and concerning upward direction has become evident in the recent past. This observation underscores the necessity to account for compound events' effects in addition to intense summer showers. The extreme flood event of July 2021, along with the observed increase in flood drivers, emphasizes the importance of incorporating compound events into flood risk assessment.

*Data availability.* Precipitation data used in this study were obtained from the Royal Netherlands Meteorological Institute (KNMI). Time series for the manual rain gauge network can be downloaded from https://www.knmi.nl/nederland-nu/klimatologie/monv/reeksen and for the meteorological station at Maastricht from https://daggegevens.knmi.nl/klimatologie/daggegevens. Discharge time series at Meerssen were provided by the water authority Waterschap Limburg and should be requested via them.

*Author contributions.* All authors contributed to the study conception and design. AT conducted all the analyses. AT prepared the manuscript, and all co-authors contributed to the content and improvement of the manuscript.

*Competing interests.* The authors declare that they have no conflict of interests.

*Acknowledgements.* This research was financially supported by the EMfloodResilience project, which was carried out within the context of Interreg V-A Euregio MeuseRhine and is partially funded by the European Regional Development Fund. The authors would like to thank the Royal Netherlands Meteorological Institute (KNMI) and the water authority Waterschap Limburg for providing all the data used in this work. We would also like to thank Deltares for supporting parts of this research.

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
