# Peer review of "Flood drivers and trends: a case study of the Geul River Catchment (the Netherlands) over the past half century"

_Hydrology and Earth System Sciences, 2023_

## Author Comment (AC1)

**Reply to RC1**

We thank the referee for his/her review and constructive comments. Original review comments are shown in **black** while our replies are provided in blue.

**This paper studies drivers of flooding and flood change in the Geul River catchment (Netherlands). Better understanding the drivers of flood change is a very topical subject and this paper produces a useful contribution on this topic. It especially stands out by providing a more in-depth insight than many large-sample studies (involving many catchments at once) have managed to provide on this topic, while still providing methodologies and insights that can be more widely adopted in understanding drivers of flood and flood change. However, at the same time, the paper seems to suffer from one major issue. I personally recommend the publication of this after this can be addressed meaningfully.**

**"Major comments"**

**The consideration of antecedent wetness as a flood driver relies on a threshold API value (exceeding 1). This API index is based on antecedent precipitation and does not take any evaporative processes into account. The latter seems somewhat problematic as soil wetness in this region tends rho be very seasonal (as ET is low in winter and high in summer) which very likely causes the strong seasonality in maximum flow and flood events (see e.g. Figure 3) but which is not visible in any of the considered flood drivers. Therefore it seems that the importance of soil wetness does not reflect soil wetness in this paper, but reflects relative wetness compared to what is normal for that part of the season (which is not relevant to the study?). This problem likely causes a strong bias in all results and thus the overall conclusions.**

**Answer:** The primary concern highlighted pertains to the exclusion of evaporative processes in the Antecedent Precipitation Index (API) utilized for evaluating initial catchment conditions. The simple question is: "Is API a reliable indicator for soil wetness for this study region year-round, or is its meaning seasonally dependent?". To investigate this, the simple 30-day before an event effective rainfall (precipitation minus potential evaporation from the Maastricht station) was calculated instead of the simple API, and Fig. 5 from the manuscript was reproduced (see Fig. R1 below), expressing $Q_{max}$ in mm/day this time.

Larger offsets are visible in the summer half-year events and the overall correlation is low (as in the initial analysis), however, the correlation between the $P_{99}$ and $P_{MD}$ events and the $Q_{max}$ (purple markers and reported correlation value) is maintained (Fig. R1), which is in line with the original analysis. Only 7/49 $Q_{max}$ events occurred in summer and 6/7 summer $Q_{max}$ events are in the $P_{99}$ and $P_{MD}$ events, so the calculated correlation includes them. In addition, the top five floods remain "higher" compared to other events. Thus, we believe that our main conclusions are not biased. We will include this analysis in the revised version of our manuscript.

[Figure]

***Figure R1*** *Annual maxima events ($Q_{max}$) and their 30-day pre-event effective rainfall at Maastricht (a) and Vaals (c), including their preceding extreme indicators. The top five floods during the study period are shown with their year of occurrence. The dashed purple line represents the linear fit, using the least squares approach, between the effective rainfall of the high flow events preceded by $P_{99}$ and $P_{MD}$ and their respective $Q_{max}$ values. The total (four-day) precipitation versus $Q_{max}$ is also presented for these events at Maastricht (b) and Vaals (d). The shaded area shows the 95% confidence intervals for the fits and the Pearson's correlation coefficients ($\rho$) are also reported.*

**"Minor comments"**

- It seems like the statement "Results suggest that extreme 24-hour precipitation cannot solely lead to floods." is unlikely but not physically impossible. Therefore, I recommend rephrasing "cannot".

- L15: "Unprecedented precipitation" seems like a bold statement when it's not specified for example since the observational record started, or some clause that determines the period over which we talk.

- L33: this statement could, in addition, be supported by some other publications that show the importance of antecedent wetness in other places.

- Fig 2. Check the label of "Feb".

- L144: "all-4day" misspelled?

- I'd recommend (but maybe this is just personal taste you can ignore) to start the results paragraph with a sentence that summarizes the result. This would make it easier for a reader to focus on when reading the details in the figure that follows. This essentially applies to each new paragraph in the results.

**Answer:** We thank the reviewer for the several minor comments. These will be addressed in a point-by-point response whilst preparing a revised version of our manuscript.

---

## Author Comment (AC2)

**Reply to RC2**

We thank the referee for his/her review and constructive comments. Original review comments are shown in **black** while our replies are provided in blue.

**The manuscript "Flood drivers and trends: a case study of the Geul River Catchment over the past half century" by Tsiokanos et al., analyses the long-term temporal variability of flood drivers for the Geul river catchment. The study adopts an interesting multi-temporal approach to analyze temporal trends of floods and their drivers and finds that 1-day extreme precipitation alone does not explain flood changes, rather heavy prolonged rain, and wet initial conditions. The manuscript is well written, and the analyses and results are presented in a convincing way. Please find my comments below:**

**"Major comment"**

1. **The aim of the study should be clarified. Is it to develop a methodology (L61-64) or to understand flood trends and their drivers in the catchment (L69-72)? These lines appear quite disconnected in the introduction. Furthermore, is the multi-temporal trend approach new (L7, L61-64), or it was proposed by Hannaford et al. (2021) and Murphy et al. (2020) as stated in line 52-53? Please clarify.**

**Answer:** The main objective of this study is to identify the primary drivers of high-flow/flood events in the Geul river catchment and examine their long-term trends. To achieve this, we employ an event-based approach (examining the relative contributions of extreme precipitation, prolonged heavy rainfall, extreme initial conditions, and compound extremes in generating high flows) and we use a multi-temporal trend analysis. Multi-temporal trend analysis is not novel, as it has been employed to detect temporal variabilities in the past. However, in this work, we build on the multi-temporal approach and propose a new methodology to assess the consistency or stability of trends in this analysis.

We aim to contribute valuable insights to the Geul area without presenting the article solely as a case study. Our combined approaches (integrating an event-based approach with multi-temporal analyses) and the proposed trend consistency methodology can be applied to diverse studies. Furthermore, in future work, we seek to extrapolate our findings to yield useful outcomes for similar regions worldwide (Section 4.2). We will further clarify the focus of the study in the revised version accordingly.

**"Specific comments"**

1. **"critical precipitation" terminology. In several parts of the manuscript (abstract, introduction and discussion) the authors draw conclusions on the "critical precipitation (precipitation that leads to floods)". It is not fully clear to what of the analyzed precipitation indices they refer to. Please clarify.**

**Answer:** With the term "critical precipitation" in the aforementioned sections we refer to the consistently increasing trends in winter. In this respect, we find that the rise in severe precipitation is mostly caused by more rain on wet days and that prolonged events have impacted the area and caused floods in winters. At the same time, we show that heavy storms in combination with wet antecedent conditions should be considered. As a result, with the term "critical precipitation" in winter, we refer to the increases in prolonged heavy events $P_{kD}$ for k > 3 days and/or $P_{MD}$ but also to the increases in daily extremes in $P_{1D}$ and $P_{95}$ as together with wet antecedent conditions they can lead to compound events and thus flooding. We will further clarify this in the revised version.

2. **L145-148: It is not clear if these lines describe an extra criterion used. How do you practically ensure that P$_{MD}$ is higher than P$_{99}$ ? What do you do when this is not the case (L148)?**

**Answer:** The $P_{MD}$ events are defined using the 95$^{th}$ percentile of all 4-day accumulated (rolling) precipitation sums and the $P_{99}$ events using the 99$^{th}$ percentile of wet days (days with more than 1 mm precipitation). For each of the five precipitation stations considered the $P_{MD}$ 95$^{th}$ percentile was calculated and it was found to be higher than the 99$^{th}$ percentile used for the definition of $P_{99}$. Line 148 refers to very extreme events when the 24-hour precipitation events can cause at the same time both $P_{99}$ and $P_{MD}$ which is unavoidable (especially in the way we defined $P_{99}$, using two days -see definition line 180-), and not in cases when the $P_{MD}$ 95$^{th}$ percentile is lower than the $P_{99}$ (which is not feasible). These lines will be reformulated to avoid confusion. Generally, this effect is taken into account in the results (see lines 256-260) and is further discussed in subsection 3.1.1, especially in lines 304 – 308.

3. **L158: how is FE defined?**

**Answer:** Past flood events (FE) come from the definition/description given earlier in subsection 2.2 (Data sets), lines 93-98. We will provide additional clarification in the text, or we may opt not to use the acronym "FE" due to the limited number of occurrences in the manuscript.

4. **L207: "Trends in P$_{kD}$ are based on the. annual maximum values". What does it mean? Do you refer to annual maximum discharges and the fact that P$_{kD}$ is calculated using k days hat preceding flood events? Please clarify.**

**Answer:** Trends in $P_{kD}$ are not connected to the occurrence of $Q_{max}$. They are calculated based on the highest k-day total precipitation per year. So actually, a summation moving window with different lengths (1, 3, 5, 7, 10, 15, 30, and 40 days) is applied over the whole time series from the 1950s to 2021 and the annual maxima (in the season half-years) are extracted. This will be also clarified in the revised version of the manuscript to avoid confusion, especially with Table 4 (correlation coefficients between the winter half-year discharge maxima and their antecedent k-day precipitation depths).

5. **L218-220: Why are different assumptions used for the MK test for precipitation and discharge trends? Why do you account for autocorrelation in annual maximum discharge series? Annual maximum values are typically considered uncorrelated by construction as they belong to different blocks/years.**

**Answer:** We used the original MK test for precipitation, assuming no (auto)correlation, because precipitation time series are considered less prone to autocorrelation due to the inherent variability in weather patterns (strong random variation in the daily time series). For discharge, we applied a modified MK test that accounts for autocorrelation to ensure the statistical robustness of trend detection (autocorrelation can impact trend detection accuracy). While it is true that annual maximum discharge values are typically considered uncorrelated by construction, low (first-order) autocorrelation might be present. In any case, the difference between the original and the modified MK for the discharge time series appears to be minor. We will reevaluate the necessity of employing a modified MK test for discharge.

6. **L221: What do you consider in the analyses?**

**Answer:** We are not entirely sure about the specific clarification the reviewer is seeking with this comment. Trends are considered statistically significant at α=0.2. The criteria used to categorize trend's consistency are described in lines 228-233. These criteria are based on the statistical significance (percentage of time *t* for which trends are statistically significant) and their directions (number of the detected statistically significant trends that are in the same direction, i.e. increasing or decreasing)

mentioned in line 221. For example, to have a consistent trend we need 25-45% of all the calculated trends in the multi-temporal analyses to be statistically significant while at the same time the majority (more than 60%) of the detected significant trends should be in the same direction (increasing or decreasing).

**7. Table 2: Last column. Shouldn't it be "Reverse *relative* frequency?**

**Answer:** The last column in Table 2 shows the relative frequencies of the reserve situation: given that a driver has occurred, what is the relative frequency of it being followed by a $Q_{max}$ event. It is again a relative frequency calculation (column (3) divided by column (2)). We will consider reformulating this to avoid confusion with Fig. 4 (relative frequencies but for a different scenario).

**8. L304-308: these lines were not fully clear to me.**

**Answer:** We define $P_{MD}$ using the 4-day accumulated precipitation amount exceeding the 95th percentile thresholds (the percentile is calculated from all 4-day accumulated rolling sums). As was explained, for very extreme 24-hour events this amount can sometimes be exceeded and can cause at the same time both $P_{MD}$ and $P_{99}$. In that case, the $P_{MD}$ is not caused by the 4-day accumulated amount but by the one-day event. However, when we are using a longer accumulation period for $P_{MD}$ (i.e. 5, 6, 7, 8, 9, and 10 days) the corresponding 95th percentile increases, as the moving/accumulated period is extended, and becomes much larger than the 99th percentile used for the definition of $P_{99}$. Thus, the separation between $P_{MD}$ and $P_{99}$ becomes clearer. Irrespective of the applied duration our results remain relatively stable.

---

## Author Comment (AC3)

**Reply to RC3**

We thank the referee for his/her review and constructive comments. Original review comments are shown in **black** while our replies are provided in blue.

**The authors present an event-based analysis of flood drivers on a 344 km2 catchment using 50 years of concurrent daily rainfall and continuous streamflow data. The main conclusions are that heavy 4-day precipitation is the primary high flow driver and this, when combined with wet antecedent conditions, provides a stronger indication of flood likelihood than extreme daily precipitation alone.**

**Overall, I think the evidence presented provides reasonable support for the conclusions drawn, but this evidence could be strengthed and clarified. The authors have selected an interesting topic and a very worthwhile case study. I have three major comments and some minor ones, as detailed below.**

**"Major comments"**

1. **The floods considered are based on a small number of factors (daily and 4-day precipitation occurrences occurring in the highest 1% and 5% of wet-day events, API, and various joint combinations). While in concept these are reasonable surrogates for the underlying flood processes of most relevance, it is a little surprising that no attempt appears to have been made to select factors of specific relevance to the catchment. For example, rather than adopt an arbitrary API, the decay factors of an API function could be fitted to the selected flood maxima and then used in the event-by-event analysis. Alternatively, a simple daily soil-moisture accounting function could be derived that implicitly allows for the influence of rainfall sequencing and evaporation; even without fitting to any observed data such an approach would appear to have greater efficacy than the adopted indicator of wetness.**

**Answer:** Regarding the API and the exclusion of evaporative processes utilized for evaluating initial catchment conditions, the simple 30-day before an event effective rainfall (precipitation minus potential evaporation from the Maastricht station) was calculated instead of the simple API, and Fig. 5 from the manuscript was reproduced. Kindly see our reply to RC1.

**Similarly, a simple correlation analysis could be used to justify the number of days adopted for the multi-day precipitation index, as at present no discussion is provided to justify the "critical" duration adopted. Such analyses would strengthen the physical reasoning used to assess the relative importance of the different flood drivers and may reveal greater insights about the nature of the interactions involved.**

**Answer:** The selection of the number of days (i.e. four) adopted for the multi-day precipitation ($P_{MD}$) index is partly discussed in lines 143-149. A four-day duration is selected considering the hydrological functioning of the catchment (L144). As we define extreme precipitation ($P_{99}$; 24-hour precipitation exceeding the 99$^{th}$ percentile of rainy days) using two days (precipitation on the same or the previous day of the flood event) for $P_{MD}$ we need more than 3 days. The 99$^{th}$ threshold for the $P_{99}$ is extracted only from wet days, while the 95$^{th}$ for the $P_{MD}$ is taken from all 4-day accumulations (rolling sum). This is done to ensure that the 95$^{th}$ percentile of multi-day rainfall is larger than the 99$^{th}$ percentile of single-day rainfall so as to make the distinction between $P_{99}$ and $P_{MD}$ clearer. This is achieved with longer accumulation durations for $P_{MD}$.

The sensitivity of $P_{MD}$ to different precipitation durations is investigated in subsection 3.1.1. The mean relative probability of $P_{MD}$ is highest at the 4-day accumulation period (confirming the usefulness of the selected period), while for the remaining indices, the results remain relatively stable regardless of

the selected duration. As we are using the 95$^{th}$ percentile of all k-day accumulated (rolling sum) precipitation to define $P_{MD}$ and we have "daily" values, usually this threshold is exceeded in prolonged events irrespective of the selected duration. This indicates that we have prolonged (multi-day) heavy events (larger than the 95$^{th}$ percentile of the selected k-day accumulations), however not so extreme as the 24-hour $P_{99}$, which helps us examine the relative contributions of extreme precipitation and prolonged heavy rainfall in generating high flows. We will consider including this analysis earlier in the revised version of the manuscript.

Overall, in the revised version of the manuscript, we aim to enhance the physical reasoning used to assess the relative importance of the employed drivers.

2. **Most of the analyses focus on the sample of events where it is known that conditions have resulted in floods. However, concentrating on the sample of 870 multi-day precipitation events (noted in Table 2) and examining the moderating factors which led to 50 annual maxima events should provide more insight about the processes leading to floods than does focusing on the much smaller sample of known flood maxima. For example, the analysis of these 870 events using similar diagnostics to that used in Fig 5 would make it clearer what combinations of factors lead to major flooding and which don't. It may be found that the combinations of conditions that are associated with floods may in some (or many) cases not lead to flooding, and this may highlight the influence of an additional factor that has not been considered. The "reverse" analysis described in the paper thus needs more focus and attention.**

**Answer:** We agree with the reviewer that it would be more thorough to expand the reverse analysis. This will be done in the revised version of the manuscript.

3. **The results are consistent with physical reasoning though in places I had to work quite hard to follow the logic of the narrative and the specific details of the results. It would thus be useful if the authors tightened up the narrative and provided additional discussion. For example:**

    1. **the information presented in Table 3 needs further explanation as the supporting discussion on this was not particularly helpful.**

**Answer:** Additional clarification will be incorporated in the revised version.

    2. **While the information presented in Figure 4 is broadly clear, I do not understand how the relative frequencies are calculated and why selected combinations of them don't add up to 100%.**

**Answer:** As reported in the figure's label, the relative frequencies are the "count of a driver leading to $Q_{max}$ in the $Q_{max}$ cases divided by the total number of cases". We simply count how many times a driver is observed in the total number of the $Q_{max}$ events (49 cases). A single $Q_{max}$ event can be preceded at the same time by more than one flood indicator, e.g. most of the $P_{99}$ events that led to $Q_{max}$ were also $P_{MD}$ events (see Fig. 5 and also our answer to the following comment).

    3. **Fig 5 provides is a useful analysis as it differentiates between floods of different magnitude, yet it is not entirely clear what the different symbols are in Figure 5 denote - they appear to differ from the indicators listed in Table 1? It would perhaps be useful to examine such correlations for all selected indicators, allowing for timing lags as needed?**

**Answer:** Fig. 5 offers important insights by showing that wetter conditions (indicated by higher API values) are expected to have different effects on high flows, particularly in compound events, while it reveals the very weak correlation between $Q_{max}$ and the event precipitation. These aspects were not as apparent in the analyses presented earlier (Fig. 3 and 4).

The symbol/marker of each $Q_{max}$ event indicates its preceding indicator as defined in Table 1 and is connected with Fig. 4, however, it should be interpreted slightly differently. In Fig. 5, we can also see the specific drivers preceding $Q_{max}$, and subsequently the overlapping mentioned in the previous reply and partly the distinction between $P_{99}$ and $P_{MD}$ (that's why we present also the "only $P_{99}$" and "only $P_{MD}$" events).

The $P_{99}$ markers (both in green and purple) in "Wet" classification (i.e., API > 1.5) indicate Compound I, the $P_{MD}$ markers (both in orange and purple) in "Wet" classification indicate Compound II, and the $P_{99}$ & $P_{MD}$ purple markers in "Wet" classification indicate Compound III. All $P_{MD}$ markers (both orange and purple markers), irrespective of their wetness (API), are classified as $P_{MD}$ and thus used to calculate the relative frequencies of $Q_{max}$ being preceded by $P_{MD}$ in Fig.4b (e.g. 37/49 * 100% = 76 % relative frequency at Maastricht station), and so on. For example, the 1970 flood event in Fig. 5a at Maastricht is classified as $P_{MD}$, $P_{WAC}$, and Compound II. In this event, no $P_{99}$ is observed. Or, for example, the 1987 event at Vaals (Fig. 5c) is classified only as $P_{WAC}$.

Correlations among all chosen indicators will be analyzed, and additional discussion will be included to provide further clarification on the meaning of Fig. 5.

4. **Further efforts should be made to strengthen the narrative thread throughout the paper as in many places I found myself going back and forth within the current and previous paragraphs to make sure I was following the intended logic. For example in Section 2.3.3 the discussion around the logic of the selected indicators commences before they are clearly defined two paragraphs later.**

**Answer:** We will work on improving the narrative flow throughout the paper and make it easier to follow.

**"Minor comments"**

- **Figure 2(b) – x-axis label is incorrect (it is not a rate, but rather the proportion of time that the given flows are exceeded)**

- **Line 219-220 – why is it the serial correlation of the precipitation time series assumed and not simply calculated?**

- **Line 249-250 – the justification for the last sentence of this paragraph is not clear**

- **Line 256 – should 86% be 83.7%?**

- **Line 397 – clearer justification is required for the 3rd sentence in this para regarding the cause for the rise in severe precipitation**

- **There are numerous small errors with the use of prepositions and other minor grammatical problems, and these should be reviewed and corrected.**

**Answer:** We thank the reviewer for the several minor comments. These will be addressed in a point-by-point response whilst preparing a revised version of our manuscript.

---

## Author Response (AR1)

April 13th, 2024
Prof. Dr. Nadav Peleg
Editor - Hydrology and Eart System Sciences (HESS)

RE: Manuscript HESS-2023-263

Dear Prof. Dr. Nadav Peleg,

We greatly appreciate the time and effort you put into handling our manuscript. We would also like to thank the three anonymous referees for their thorough review and constructive feedback, which have proven valuable in enhancing the quality of our work. Enclosed, please find the revised version of the manuscript along with a separate document detailing the tracked changes made. Furthermore, we have included below point-by-point responses to each referee's comments as requested. The original review comments are presented in black, while our responses are highlighted in blue. Please note that line numbers, sections, and figure numbers in our responses are based on the revised version of the manuscript.

Sincerely,

Athanasios Tsiokanos

On behalf of Martine Rutten, Ruud J. van der Ent, and Remko Uijlenhoet

**# Reply to RC1**

This paper studies drivers of flooding and flood change in the Geul River catchment (Netherlands). Better understanding the drivers of flood change is a very topical subject and this paper produces a useful contribution on this topic. It especially stands out by providing a more in-depth insight than many large-sample studies (involving many catchments at once) have managed to provide on this topic, while still providing methodologies and insights that can be more widely adopted in understanding drivers of flood and flood change. However, at the same time, the paper seems to suffer from one major issue. I personally recommend the publication of this after this can be addressed meaningfully.

"Major comments"

The consideration of antecedent wetness as a flood driver relies on a threshold API value (exceeding 1). This API index is based on antecedent precipitation and does not take any evaporative processes into account. The latter seems somewhat problematic as soil wetness in this region tends rho be very seasonal (as ET is low in winter and high in summer) which very likely causes the strong seasonality in maximum flow and flood events (see e.g. Figure 3) but which is not visible in any of the considered flood drivers. Therefore it seems that the importance of soil wetness does not reflect soil wetness in this paper, but reflects relative wetness compared to what is normal for that part of the season (which is not relevant to the study?). This problem likely causes a strong bias in all results and thus the overall conclusions.

**Answer:** The primary concern highlighted pertains to the exclusion of evaporative processes in the Antecedent Precipitation Index (API) utilized for evaluating initial catchment conditions. The simple question is: "Is API a reliable indicator for soil wetness for this study region year-round, or is its meaning seasonally dependent?". To investigate this, the simple 30-day before an event effective rainfall (precipitation minus reference evaporation) was calculated instead of the simple API, and Fig. 5 from the manuscript was reproduced expressing $Q_{max}$ in mm/day this time.

This new analysis was documented in a separate subsection in the results (Sect. 3.1.5). In addition, the following was added to the methodology L168-172: "The API's effectiveness in assessing initial soil wetness conditions was documented for instance by Marchi et al. (2010), who demonstrated its strong agreement with predictions from a continuous soil moisture accounting hydrological model (Norbiato et al., 2008). However, since the index is based solely on precipitation, its sensitivity to evaporation is further discussed in Sect. 3.1.5. This is done by computing the 30-day pre-event effective rainfall, which entails subtracting reference evaporation obtained from the Maastricht station from the precipitation measurements.".

"Minor comments"

- It seems like the statement "Results suggest that extreme 24-hour precipitation cannot solely lead to floods." is unlikely but not physically impossible. Therefore, I recommend rephrasing "cannot".

**Answer:** The aforementioned statement was rephrased to the following: "Results suggest that extreme 24-hour precipitation alone is typically insufficient to cause floods."

- L15: "Unprecedented precipitation" seems like a bold statement when it's not specified for example since the observational record started, or some clause that determines the period over which we talk.

**Answer:** Changed to "extreme precipitation".

- L33: this statement could, in addition, be supported by some other publications that show the importance of antecedent wetness in other places.

**Answer:** Additional literature was added to support the aforementioned statement.

- Fig 2. Check the label of "Feb".

**Answer:** The label of month February was corrected.

- L144: "all-4day" misspelled?

**Answer:** changed to "all $k$-day".

- I'd recommend (but maybe this is just personal taste you can ignore) to start the results paragraph with a sentence that summarizes the result. This would make it easier for a reader to focus on when reading the details in the figure that follows. This essentially applies to each new paragraph in the results.

**Answer:** A few paragraphs (and sentences) in the result sections were modified to incorporate the suggested writing style, for example in the following sections: Section 3.1.1, Section 3.1.2, Section 3.1.4, and Section 3.1.5

**# Reply to RC2**

The manuscript "Flood drivers and trends: a case study of the Geul River Catchment over the past half century" by Tsiokanos et al., analyses the long-term temporal variability of flood drivers for the Geul river catchment. The study adopts an interesting multi-temporal approach to analyze temporal trends of floods and their drivers and finds that 1-day extreme precipitation alone does not explain flood changes, rather heavy prolonged rain, and wet initial conditions. The manuscript is well written, and the analyses and results are presented in a convincing way. Please find my comments below:

"Major comment"

1. The aim of the study should be clarified. Is it to develop a methodology (L61-64) or to understand flood trends and their drivers in the catchment (L69-72)? These lines appear quite disconnected in the introduction. Furthermore, is the multi-temporal trend approach new (L7, L61-64), or it was proposed by Hannaford et al. (2021) and Murphy et al. (2020) as stated in line 52-53? Please clarify.

**Answer:** The main objective of this study is to identify the primary drivers of high-flow/flood events in the Geul river catchment and examine their long-term trends. To achieve this, we employ an event-based approach (examining the relative contributions of extreme precipitation, prolonged heavy rainfall, extreme initial conditions, and compound extremes in generating high flows) and we use a multi-temporal trend analysis. We aim to contribute valuable insights to the Geul area without presenting the article solely as a case study. Our combined approaches (integrating an event-based approach with multi-temporal analyses) and the proposed trend consistency methodology can be applied to diverse studies. Furthermore, in future work, we seek to extrapolate our findings to yield useful outcomes for similar regions worldwide.

Multi-temporal trend analysis is not novel, as it has been employed to detect temporal variabilities in the past. However, in this work, we build on the multi-temporal approach and propose a new methodology to assess the consistency or stability of trends in this analysis. In the aforementioned

lines, it is reported that we used a multi-temporal trend analysis to investigate temporal variabilities and a new methodology to detect the dominant direction of a trend. So these lines mention that the new methodology is related to the dominant direction (i.e. consistency) of a trend within a multi-temporal analysis and not to the multi-temporal analysis itself.

In this respect, the following texts were added/modified in the last two paragraphs of the introduction:

*L62-65* "To address these limitations, our study builds on the multi-temporal approach and develops a methodology capable of identifying and assessing trend consistency in multi-temporal analyses, taking into account the complete range of variability. This new method is anticipated to deepen our understanding of flood driver trends in the Geul River catchment, with potential applicability across broader contexts."

*L70-72* "Therefore, our objective is to detect the primary drivers of high-flow/flood events in the Geul River catchment and analyze their long-term trends. To achieve these objectives, we address the following scientific questions that are crucial for our understanding of floods in the Geul River catchment:"

*L76-79* "Although our study focuses on the Geul area, it is essential to highlight that our combined approaches (integrating an event-based approach with multi-temporal analyses) and proposed trend consistency method hold applicability beyond this specific case. Thus, our aim is to offer valuable insights for the Geul area while avoiding constraining the scope of our methods and findings to a singular case study."

"Specific comments"

1. "critical precipitation" terminology. In several parts of the manuscript (abstract, introduction and discussion) the authors draw conclusions on the "critical precipitation (precipitation that leads to floods)". It is not fully clear to what of the analyzed precipitation indices they refer to. Please clarify.

**Answer:** "critical precipitation" changed to "flood producing precipitation" in the abstract, introduction, and conclusions.

In addition, the following text was added in the discussion *L476-482*: "During this period, various indices representing heavy prolonged events such as $P_{kD}$ for k ≥ 3 days and/or $P_{MD}$, as well as 24-hour extreme indices like $P_{1D}$ and $P_{95}$, show mostly consistent increases. These combinations of indicators can contribute to the saturation of the catchment, thereby increasing the risk of flooding. In addition, it appears that a portion of the rise in severe precipitation stems from increased rainfall on already wet days, as evidenced by consistent (or strongly consistent) rises in $P_{3D}$, $P_{5D}$, $P_{7D}$, and $P_{MD}$ across the majority of stations. All these findings are crucial as heavy and prolonged storms in combination with wet antecedent conditions have impacted the catchment and caused floods mainly in winter."

2. L145-148: It is not clear if these lines describe an extra criterion used. How do you practically ensure that $P_{MD}$ is higher than $P_{99}$ ? What do you do when this is not the case (L148)?

**Answer:** The $P_{MD}$ events were defined using the 95$^{th}$ percentile of all 4-day accumulated (rolling) precipitation sums and the $P_{99}$ events using the 99$^{th}$ percentile of wet days (days with more than 1 mm precipitation). For each of the five precipitation stations considered the $P_{MD}$ 95$^{th}$ percentile was calculated and it was found to be higher than the 99$^{th}$ percentile used for the definition of $P_{99}$. Previous line 148 referred to very extreme events when the 24-hour precipitation events can cause at the same

time both $P_{99}$ and $P_{MD}$ which is unavoidable (especially in the way we defined $P_{99}$, using two days) and not in cases when the $P_{MD}$ 95$^{th}$ percentile is lower than the $P_{99}$ (which is not feasible).

The $P_{MD}$ definition (**L150-163**) in the Extreme Indicators section was adjusted to the following (also in line with RC3 major comment 1): "We define $P_{MD}$ events using the 95th percentile of all $k$-day accumulated (rolling sum) precipitation time series (Nanditha and Mishra, 2022). To clarify the $P_{MD}$ definition, we ensure that the 95th percentile of multi-day rainfall consistently surpasses the 99th percentile of the 24-hour rainfall on wet days, aiding in distinguishing between $P_{99}$ and $P_{MD}$. In this way usually more than two days of precipitation are necessary to exceed the k-day 95th percentile and trigger $P_{MD}$, allowing the assumption that $P_{MD}$ can be used as a proxy of heavy prolonged rainfall. As we use the 95th percentile of all k-day accumulated (rolling sum) precipitation to define $P_{MD}$ and we have "daily" values, this threshold is expected to be exceeded in prolonged events irrespective of the selected duration, indicating that we have prolonged (multi-day) heavy events (larger than the 95th percentile of the selected k-day accumulations), although not so extreme as the 24-hour $P_{99}$ , which helps us examine the relative contributions of extreme precipitation and prolonged heavy rainfall in generating high flows. However, in extremely rare cases, 24-hour precipitation can simultaneously trigger both $P_{99}$ and $P_{MD}$ for the lower k-day accumulation periods, which is unavoidable. Thus, for each of the five precipitation stations considered, we calculated the $P_{MD}$ 95th percentile for different durations. It was found that a duration longer than 4 days is required for this percentile to surpass the 99th percentile used in defining $P_{99}$. Finally, to determine the most suitable k-day $P_{MD}$ duration for k ≥ 4, we evaluate the $P_{MD}$ probability preceding high flows across multi-day precipitation durations up to 10 days (see Sec. 3.1.1)."

3. L158: how is FE defined?

**Answer:** The acronym "FE", which denoted past flood events as defined in subsection 2.2 (Data sets) is not used anymore in the manuscript due to the limited number of occurrences to avoid confusion. Additional clarification was added in subsection 2.3.2 "Monthly distribution of extremes" and in Fig. 3 label "FE" was replaced by "Floods".

4. L207: "Trends in $P_{kD}$ are based on the. annual maximum values". What does it mean? Do you refer to annual maximum discharges and the fact that $P_{kD}$ is calculated using k days hat preceding flood events? Please clarify.

**Answer:** Trends in $P_{kD}$ are not connected to the occurrence of $Q_{max}$. They are calculated based on the highest k-day total precipitation per year. The following clarification was added (**L233-234**): "Trends in $P_{kD}$ are based on the highest $k$-day total precipitation per year (a summation moving window with different lengths is applied over the whole time series from the 1950s to 2021 and the annual maxima are extracted)".

5. L218-220: Why are different assumptions used for the MK test for precipitation and discharge trends? Why do you account for autocorrelation in annual maximum discharge series? Annual maximum values are typically considered uncorrelated by construction as they belong to different blocks/years.

**Answer:** The original MK test was applied for the discharge time series and the corresponding figures (i.e. Fig. 10) were updated. In addition, **L245-248** in the methods section were adjusted as follows: "The original M-K test is employed on both the precipitation indices and discharge time series, instead of a modified M-K version that accounts for the influence of serial correlation on trend calculations. This choice is guided by the assumption that the precipitation time series exhibit no significant serial correlation and that the annual maximum discharge values are typically considered uncorrelated by

construction." Overall, the differences appear to be minor. The main difference between the modified and original MK tests is the slightly lower values in the winter half-year (see Figure R1). For these reasons "Strong increasing" in **L420** was changed to "Increasing" and "the increase in the $Q_{W,max}$ is consistent" in **L422** to "the increase in the $Q_{W,max}$ is considered consistent taking also into account the missing hydrological years of 1971, 1974 and 1990".

[Figure]

*Figure R1 Comparison between trends in maxima half-year discharge values between the modified (upper panels) and the original (lower panels) MK tests.*

6.  L221: What do you consider in the analyses?

**Answer:** We are not entirely sure about the specific clarification the reviewer is seeking with this comment. Trends are considered statistically significant at α=0.2. The criteria used to categorize trend's consistency are described in **L256-261**. These criteria are based on the statistical significance (percentage of time *t* for which trends are statistically significant) and their directions (number of the detected statistically significant trends that are in the same direction, i.e. increasing or decreasing) mentioned in **L249**. For example, to have a consistent trend we need 25-45% of all the calculated trends in the multi-temporal analyses to be statistically significant while at the same time the majority (more than 60%) of the detected significant trends should be in the same direction (increasing or decreasing).

7.  Table 2: Last column. Shouldn't it be "Reverse *relative* frequency?

**Answer:** changed to "Relative (reverse) frequency".

8.  L304-308: these lines were not fully clear to me.

**Answer:** The following text was added (**L336-344**): "In the way we defined $P_{99}$ (two days interval) we observe that the $P_{99}$ events preceding a $Q_{max}$ usually coincide with also $P_{MD}$. For very extreme 24-hour

events the 4-day 95th percentile used for the $P_{MD}$ definition can be exceeded and cause at the same time both $P_{MD}$ and $P_{99}$, which is unavoidable. However, in longer accumulation periods for $P_{MD}$ (i.e. 5, 6, 7, 8, 9, and 10 days) the corresponding 95th percentile increases, as the moving/accumulated period is extended, and becomes much larger than the 99th percentile used for the definition of $P_{99}$. In these cases, irrespective of the duration the mean relative frequencies of high flows preceded by Compound II and III remain stable (see supplementary material for the analysis). This implies that preceding $P_{99}$, rainfall events (whether heavy or not) probably occurred for these events as well (at least for less extreme ones), potentially resulting in wet conditions and consequently high discharges, highlighting the correlation among the used different drivers and how they can be converted to compounds. Thus, while it is found that $Q$max is preceded by $P_{MD}$ 75% of the time, some of the $P_{MD}$ events could be forced or even caused by $P_{99}$. However, the definition of $P_{MD}$ still holds significance as it denotes an extended period of heavy rainfall."

**# Reply to RC3**

The authors present an event-based analysis of flood drivers on a 344 km2 catchment using 50 years of concurrent daily rainfall and continuous streamflow data. The main conclusions are that heavy 4-day precipitation is the primary high flow driver and this, when combined with wet antecedent conditions, provides a stronger indication of flood likelihood than extreme daily precipitation alone.

Overall, I think the evidence presented provides reasonable support for the conclusions drawn, but this evidence could be strengthed and clarified. The authors have selected an interesting topic and a very worthwhile case study. I have three major comments and some minor ones, as detailed below.

"Major comments"

1. The floods considered are based on a small number of factors (daily and 4-day precipitation occurrences occurring in the highest 1% and 5% of wet-day events, API, and various joint combinations). While in concept these are reasonable surrogates for the underlying flood processes of most relevance, it is a little surprising that no attempt appears to have been made to select factors of specific relevance to the catchment. For example, rather than adopt an arbitrary API, the decay factors of an API function could be fitted to the selected flood maxima and then used in the event-by-event analysis. Alternatively, a simple daily soil-moisture accounting function could be derived that implicitly allows for the influence of rainfall sequencing and evaporation; even without fitting to any observed data such an approach would appear to have greater efficacy than the adopted indicator of wetness.

**Answer:** Regarding the API and the exclusion of evaporative processes utilized for evaluating initial catchment conditions, the simple 30-day before an event effective rainfall (precipitation minus reference evaporation from the Maastricht station) was calculated instead of the simple API, and Fig. 5 from the manuscript was reproduced. Kindly see our reply to RC1.

Similarly, a simple correlation analysis could be used to justify the number of days adopted for the multi-day precipitation index, as at present no discussion is provided to justify the "critical" duration adopted. Such analyses would strengthen the physical reasoning used to assess the relative importance of the different flood drivers and may reveal greater insights about the nature of the interactions involved.

**Answer:** We have now included Sect. 3.1.1 "Selection of $P_{MD}$ duration" (part of the previous "Sensitivity of $P_{MD}$ to precipitation duration") in which we investigate the most appropriate accumulation period for the $P_{MD}$ definition. After this analysis, a 4-day duration is used for $P_{MD}$ throughout the remainder of

the manuscript. Also in the methods sections, additional text and explanations were added related to the critical durations $P_{MD}$ (defining for example the lower limit of the investigated durations).

2.  Most of the analyses focus on the sample of events where it is known that conditions have resulted in floods. However, concentrating on the sample of 870 multi-day precipitation events (noted in Table 2) and examining the moderating factors which led to 50 annual maxima events should provide more insight about the processes leading to floods than does focusing on the much smaller sample of known flood maxima. For example, the analysis of these 870 events using similar diagnostics to that used in Fig 5 would make it clearer what combinations of factors lead to major flooding and which don't. It may be found that the combinations of conditions that are associated with floods may in some (or many) cases not lead to flooding, and this may highlight the influence of an additional factor that has not been considered. The "reverse" analysis described in the paper thus needs more focus and attention.

**Answer:** In the revised version of the manuscript the reverse analysis was expanded, and thus more attention is given to it, as it forms Section 3.1.4 "Extreme precipitation based analysis". In addition to Table 5, the empirical CDF for the discharges caused by indicator per station is shown, providing better insights and understanding of the used extreme precipitation indices and their effect on high flows and specifically flood events.

3.  The results are consistent with physical reasoning though in places I had to work quite hard to follow the logic of the narrative and the specific details of the results. It would thus be useful if the authors tightened up the narrative and provided additional discussion. For example:

    1.  the information presented in Table 3 needs further explanation as the supporting discussion on this was not particularly helpful.

**Answer:** The following text was added ***L325-330***: "Examining the preceding conditions for the major past floods it appears that in most of these cases, while the precipitation events spanning 1 to 3 days were heavy, the overall precipitation over the 30 days preceding the events was substantial. This extended period of precipitation likely played a critical role in saturating the catchment, making it more susceptible to flooding. The combination of intense rainfall over shorter durations and continuous precipitation over the 30-day period seemed to collectively contribute to the formation of wet initial conditions, ultimately increasing the risk and eventually resulting in flooding"

    2.  While the information presented in Figure 4 is broadly clear, I do not understand how the relative frequencies are calculated and why selected combinations of them don't add up to 100%.

**Answer:** As reported in the figure's label, the relative frequencies are the "count of a driver leading to $Q_{max}$ in the $Q_{max}$ cases divided by the total number of cases". We simply count how many times a driver is observed in the total number of the $Q_{max}$ events (49 cases). A single $Q_{max}$ event can be preceded at the same time by more than one flood indicator, e.g. most of the $P_{99}$ events that led to $Q_{max}$ were also $P_{MD}$ events. This should have been clarified now based on the overall previous additions.

    3.  Fig 5 provides is a useful analysis as it differentiates between floods of different magnitude, yet it is not entirely clear what the different symbols are in Figure 5 denote - they appear to differ from the indicators listed in Table 1? It would perhaps be useful to examine such correlations for all selected indicators, allowing for timing lags as needed?

**Answer:** Fig. 5 was updated to include the latest definition of Compound I (and III) and the following additional paragraph was added:

"Figure 5 shows the $Q_{max}$ events plotted against their API, including also their preceding precipitation indicators (i.e. $P_{99}$ and $P_{MD}$) at the Maastricht and Vaals stations. This figure actually presents how the different events are classified based on the preceded defined indicators (Table 1), emphasizing the influence of wet conditions on high flows and exploring correlations between $Q_{max}$ and associated precipitation amounts ($P_{99}$ or $P_{MD}$). For example, all $P_{MD}$ markers (both orange and purple markers), irrespective of their wetness (API), are classified as $P_{MD}$ and thus used to calculate the relative frequencies of $Q_{max}$ being preceded by $P_{MD}$ in Fig.4b. Furthermore, the figure reveals overlapping event classifications, where one event can align with multiple indicators at the same time, e.g. a Compound III event is classified as $P_{WAC}$, $P_{99}$, $P_{MD}$, Compound I and II, while a $P_{99}$ event can be at the same time $P_{MD}$. A $Q_{max}$ event preceded by $P_{99}$ may appear in the Wet classification without being classified as Compound I. This is because we require that $P_{99}$ should occur under existing $P_{WAC}$ conditions to be classified as Compound I. This condition is imposed to prevent a $P_{99}$ event from inflating the API the day before the event, potentially leading to an API > 1.5 on the day of the event (see Sect. 2.3). For example, the 1970 event at Vaals (Fig. 5d), which is preceded by $P_{99}$, $P_{MD}$, $P_{WAC}$, and Compound II, is not classified as Compound I or III."

Also, the label of the figure was updated as follows:

"Annual maxima events ($Q_{max}$) and their Antecedent Precipitation Indices (API) at Maastricht (a) and Vaals (d), including their preceding extreme indicators. Orange markers denote events preceded solely by $P_{MD}$, green markers indicate events preceded exclusively by $P_{99}$, purple markers represent events preceded by both $P_{99}$ and $P_{MD}$ (thus classified as both $P_{99}$ and $P_{MD}$ in Fig. 4), and blue markers signify events without any extreme precipitation indicator preceding them. Purple and orange markers within the "Wet" classification, along with $P_{WAC}$, $P_{99}$, and/or $P_{MD}$ classifications, are also classified as Compound II events in Fig. 4. Discharge events preceded by Compound III (and thus Compound I and II) are indicated with red circles. The top five floods during the study period are shown with their year of occurrence. The dashed purple line represents the linear fit, using the least squares approach, between the API of the high flow events preceded by $P_{99}$ and $P_{MD}$ and their respective $Q_{max}$ values, while the red dashed line represents the linear fit between the API of the Compound III events and their $Q_{max}$. The total four-day precipitation versus $Q$max is presented for these events at Maastricht (b) and Vaals (e), and also the highest 24-hour precipitation (highest of the two $P_{99}$ amounts on the day of the event or the previous day) versus $Q_{max}$ at Maastricht (c) and Vaals (f). The shaded area shows the 95% confidence intervals for the fits, and the Pearson's correlation coefficients ($\rho$) are also reported."

4. Further efforts should be made to strengthen the narrative thread throughout the paper as in many places I found myself going back and forth within the current and previous paragraphs to make sure I was following the intended logic. For example in Section 2.3.3 the discussion around the logic of the selected indicators commences before they are clearly defined two paragraphs later.

**Answer:** Many adjustments were made to further improve the narrative flow throughout the paper and make it easier to follow, including adjustments concerning RC1 and RC2 comments. Among others:

Transferring segments of the discussion concerning the rationale behind the chosen indicators from section 2.3.3 to section 2.3.1 In **L147-148**, it is now stated, "These indicators allow us to examine the relative role of extreme precipitation, prolonged heavy rainfall, extreme soil moisture conditions, and compound extremes in generating high flows."

Additional explanation was added to 2.3.3 to further describe how the six indicators were calculated in the $Q_{max}$ events. Specifically now in **L199-206**, it is mentioned: "For Compound I events ($P_{99}$ on $P_{WAC}$), we verify whether the $P_{99}$ on $P_{WAC}$ occurred either on the event day itself or if the $P_{99}$ on $P_{WAC}$ took place one day prior to the event. This approach guarantees that the $P_{99}$ consistently occurs in pre-existing wet conditions. Thus, we establish the requirement for a $P_{99}$ to appear on $P_{WAC}$ for a compound event, thereby preventing scenarios where a $P_{99}$ occurring one day prior to the $Q_{max}$ under normal circumstances could increase the API on the day of the $Q_{max}$, leading to a $P_{WAC}$. In a similar manner, we calculate the probability of Compound III preceding the $Q{max}$ events: $P_{MD}$ and $P_{99}$ and $P_{WAC}$ on the day of the event or one day before. For the remaining indices, $P_{MD}$ and $P_{WAC}$, we simply verify whether these indicators are present on the day of the event."

"Minor comments"

- Figure 2(b) – x-axis label is incorrect (it is not a rate, but rather the proportion of time that the given flows are exceeded)

**Answer:** The x-axis label was changed to "Exceedance time".

- Line 219-220 – why is it the serial correlation of the precipitation time series assumed and not simply calculated?

**Answer:** Trends in the precipitation indices are calculated per half-year period and thus, considering also the strong variations in daily precipitation time series, it is deemed unnecessary to calculate correlations.

- Line 249-250 – the justification for the last sentence of this paragraph is not clear

**Answer:** The following lines were added at the end of the paragraph: "Although extreme precipitation events tend to occur more frequently during the summer months, high flow and flood events do not align with these periods. Factors such as antecedent soil moisture conditions, as well as the timing, duration, and intensity of rainfall events, may exert a more significant influence on high-flow generation in the catchment. Therefore, greater attention is required in understanding these factors.".

Also now the paragraph starts by summarizing the results: "In the analysis of the seasonal distribution of extreme precipitation, high flow events, and flood drivers, the findings indicate that extreme precipitation, although more frequent during summer months, does not consistently coincide with high flow events".

- Line 256 – should 86% be 83.7%?

**Answer:** Correct. Changed to "approximately 84%".

- Line 397 – clearer justification is required for the 3rd sentence in this para regarding the cause for the rise in severe precipitation

**Answer:** This point was covered. Kindly see our response to RC2 specific comment 1.

- There are numerous small errors with the use of prepositions and other minor grammatical problems, and these should be reviewed and corrected.

**Answer:** Several corrections have been made to address preposition mistakes and other minor grammatical problems.